

# Glacio-hydrological melt and runoff modelling: a limits of acceptability framework for model selection

Jonathan D Mackay[1,2], Nicholas E Barrand[1], David M Hannah[1], Stefan Krause[1], Christopher R Jackson[2], Jez Everest[3], and Guðfinna Aðalgeirsdóttir[4]

[1]School of Geography, Earth and Environmental Sciences, University of Birmingham, Edgbaston, Birmingham, B15 2TT, UK
[2]British Geological Survey, Environmental Science Centre, Keyworth, Nottingham, NG12 5GG, UK
[3]British Geological Survey, Lyell Centre, Research Avenue South, Edinburgh, EH14 4AS, UK
[4]Institute of Earth Sciences, University of Iceland, 101 Reykjavík, Iceland

*Correspondence to:* Jonathan D Mackay (joncka@bgs.ac.uk)

**Abstract.** Glacio-hydrological models (GHMs) underpin our understanding how future climate change will affect river flow regimes in glaciated watersheds. A variety of simplified GHM structures and parameterisations exist, yet the performance of these are rarely quantified at the process-level or with metrics beyond global summary statistics. A fuller understanding of the deficiencies in competing model structures and parameterisations and the ability of models to simulate physical processes require performance metrics utilising the full range of uncertainty information within input observations. Here, the glacio-hydrological characteristics of the Virkisá river basin in southern Iceland are characterised using 33 'signatures' derived from observations of ice melt, snow coverage and river discharge. The uncertainty of each set of observations are harnessed to define 'limits of acceptability' (LOA), a set of criteria used to objectively evaluate the acceptability of different GHM structures and parameterisations. This framework is used to compare and diagnose deficiencies in three melt and three runoff-routing model structures. Increased model complexity is shown to improve acceptability when evaluated against specific signatures, but does not always result in better consistency across all signatures, emphasising the difficulty in appropriate model selection and the need for multi-model prediction approaches to account for model selection uncertainty. Melt and runoff-routing structures demonstrate a hierarchy of influence on river discharge signatures with melt model structure having the most influence on discharge hydrograph seasonality and runoff-routing structure on shorter-timescale discharge events. None of the tested GHM structural configurations returned acceptable simulations across the full population of signatures. The framework outlined here provides a comprehensive and rigorous assessment tool for evaluating the acceptability of different GHM process hypotheses. Future melt and runoff model forecasts should seek to diagnose structural model deficiencies and evaluate diagnostic signatures of system behaviour using the LOA framework.

## 1 Introduction

Computational GHMs underpin our current understanding of how future climate change will affect river flow regimes in glaciated watersheds (Ragettli et al., 2016; Singh et al., 2016; Teutschbein et al., 2015; Lutz et al., 2014; Radić and Hock, 2014). A variety of GHM codes exist (e.g. Bergström, 1997; Ciarapica and Todini, 2002; Schulla, 2015; Huss et al., 2008b; Boscarello



et al., 2014; Schaefli et al., 2014), each of which include a number of model components that represent two broad groups of processes: i) glaciological mass balance: the accumulation and ablation of snow and ice; and ii) hydrological water balance: the storage and release of melt and rainfall through snow, ice, overland and the subsurface. The exact form that these model components should take, both in terms of their governing equations (structure) and numerical constants (parameterisation)

is not known. Physically-based models which solve equations derived from first principles, typically over a distributed grid, are our closest approximation of the 'true' structure. However, limited parameterisation data and computer resources often preclude the use of such complex models, particularly in remote mountainous regions where data are scarce and where the inclusion of extra complexity does not guarantee better predictions (e.g. Gabbi et al., 2014).

Simplified process models offer an alternative. They are faster to run and employ fewer parameters that are typically cal-
ibrated to available observation data. They are based on, but do not necessarily adhere to physical laws and as such their mathematical structure is somewhat unconstrained and may be biased towards a particular scientist's own perceptions and understanding of environmental processes. This has led to the development of a variety of competing model structures which purport to simulate the same process, but which have been derived from different process hypotheses. For example, a number of simplified 'index' model structures of snow and ice melt exist. The classical temperature index model (TIM) simulates melt as
a linear piecewise function of temperature only (Braithwaite, 1995), a hypothesis that can be justified because of the influence temperature has on the total energy balance of ice and snow, particularly in temperate climates (Aðalgeirsdóttir et al., 2011; Guðmundsson et al., 2009; Ohmura, 2001). So-called 'enhanced' TIM structures have also been proposed which include added levels of complexity with the purpose of providing more accurate estimates of melt. These have accounted for perturbations in melt caused by topographic shading (Hock, 1999), surface albedo characteristics (Pellicciotti et al., 2005; Oerlemans, 2001)
and debris cover (Carenzo et al., 2016).

Similarly a number of simplified representations of processes governing the hydrological water balance have been used in GHMs. Arguably, the equations that govern the routing (transport) of runoff are most important in relation to river flow predictions in glaciated river basins, as storage characteristics of ice and snow strongly influence river flow regimes over a range of time-scales (Jansson et al., 2003). The concept of linear reservoirs is the most widely adopted simplified approach for
runoff-routing in glaciated basins (Zhang et al., 2015; Hanzer et al., 2016; Gao et al., 2017). A linear reservoir lumps all of the interacting, non-linear and non-stationary components of water transmission within a pre-defined area (e.g. a watershed) into a single 'leaky bucket'. Despite its simplicity, the linear reservoir has shown to be remarkably versatile at capturing the storage-discharge characteristics of glaciated river basins around the world (Hock and Jansson, 2005; de Woul et al., 2006; Farinotti et al., 2012). This is partly because the concept lends itself to structural modifications that can represent different
glacio-hydrological systems. Hanzer et al. (2016) hypothesised that the snow pack, firn layer, glacier ice and the region free from ice all exhibit unique runoff-discharge responses and advocate the use of four linear reservoirs in parallel to distinguish between these units. However, simpler structural configurations using only two linear reservoirs in parallel to route meltwater through the snowpack and ice separately (Hannah and Gurnell, 2001) or even a single linear reservoir to route all rainfall and melt runoff simultaneously (Boscarello et al., 2014) can accurately reproduce river discharge time-series.





The availability of multiple, presumably plausible, simplified model structures presents somewhat of a dilemma to glaciologists and hydrologists as they are left with some uncertainty about how processes should be represented in their models. For the purpose of river discharge predictions, this problem is particularly pertinent as there are competing structures for two fundamental controls on these predictions: snow and ice melt and runoff-routing. One approach to mitigate this is to determine

the 'optimum' structure that best captures the observation data. Structural optimisation of simplified runoff-routing routines has largely been ignored in glacio-hydrological contexts (see Hannah and Gurnell, 2001, for one notable exception), but more studies have sought to optimise and compare simplified models of melt. Gabbi et al. (2014) applied four different TIMs to Rhonegletscher, Switzerland. They found that all achieved a similar goodness-of-fit to six years of ablation stake data, but that the inclusion of a solar radiation term provided the most accurate predictions of multi-decadal measurements of ice volume

change. Irvine-Fynn et al. (2014) applied six different TIMs to the High-Arctic Midtre Lovénbreen glacier but only found minor improvements at capturing seasonal ablation stake data when various levels of complexity were introduced to the classical (temperature-only) TIM. More recently, a comparison of four TIMs applied to four glaciers in the French Alps by Reveillet et al. (2017) found no clear evidence that using an enhanced TIM over the classical temperature-only approach provided better simulations when compared to a 17-year dataset of ablation stake measurements. Mosier et al. (2016) used a multi-criterion

evaluation approach to compare the performance of different conceptual melt model structures. They compared seven competing melt model structures in two glaciated catchments in Alaska to ablation stake, river discharge and remotely-sensed snow coverage data. They found that no single model was best across all of the observation datasets, but the inclusion of a snow cold content representation consistently produced the best goodness-of-fit scores over the evaluation data.

Clearly, while some studies have provided useful insight into the comparative behaviour between competing conceptual

process hypotheses (particularly for melt), none provide any definitive reasoning for adopting (or not) a particular model structure. Of course, discriminating between competing model structures in this way is made difficult by the fact that observation data used to drive and evaluate models are uncertain and therefore, we cannot be sure whether model deficiencies represent inadequacies in the model or the data (Beven, 2016). Beven (2006) argues that because of this uncertainty and because of the fact that all models are by definition, imperfect, no one optimum model structure (or parameterisation) exists. Instead,

there is an equifinality of 'behavioural' models that make predictions within some pre-defined acceptability bounds around the observation data that take account of the various sources of modelling uncertainty. Indeed, parameter equifinality is a well recognised phenomenon in conceptual models of snow and ice melt (Gabbi et al., 2014; Jost et al., 2012; Finger et al., 2015; Pellicciotti et al., 2012; Reveillet et al., 2017). If we accept this, then a priority within the glacio-hydrological modelling community should be to establish frameworks that allow us to robustly evaluate model appropriateness and distinguish between

behavioural (acceptable) and non-behavioural (unacceptable) structures and parameterisations. Constraining the range of acceptable models is particularly important for glacio-hydrological modelling as it has been shown that model uncertainty can lead to high uncertainty in 21st century predictions of river flows in glaciated basins (Huss et al., 2014).

One potential source for inspiration is the hydrological rainfall-runoff modelling community. Their heavy reliance on an ever-expanding choice of conceptual hydrological process models to make river flow predictions prompted Gupta et al. (2008) to

discuss the need for a better framework in which to discriminate between these competing process hypotheses. They focussed





on the evaluation metrics and noted that there was an over-reliance on metrics that quantify the average performance of a model (e.g root mean squared error and Nash-Sutcliffe efficiency) which reduce information held in observation data down to a single summary statistic. They argue for a multi-criterion, 'diagnostic' approach where more of the relevant information from observation data is extracted so that inadequacies in model structures and parameterisations can be better identified.

Indeed, hydrologists are now moving away from traditional metrics of model performance in favour of more diagnostic 'signatures' of hydrological behaviour. These have typically been derived from river flow time-series and may be as simple as the mean flow (an indicator of water balance) or they can be used to characterise the distribution (e.g. flow percentiles) and the timing (e.g. autocorrelation) of flows. They have shown to have more discrimination power than traditional error metrics (Hrachowitz et al., 2014; Shafii and Tolson, 2015; Euser et al., 2013; Schaefli, 2016) and, importantly, it is also

possible to take account of their information content (i.e. their uncertainty) so that decisions about model appropriateness can be made within the uncertainties of observation data used to evaluate the model. Such an approach was first proposed by Beven (2006), where observation data uncertainty could be used to define quantitative 'limits of acceptability' (LOA) around model evaluation metrics. Using signatures as the basis for model evaluation, different model structures and parameterisations can then be systematically evaluated for their ability to capture the signatures within their LOA, allowing the modeller to

objectively diagnose model deficiencies and make decisions about model appropriateness. The LOA framework has been used to constrain the parameters of a distributed hydrological model for flood prediction (Blazkova and Beven, 2009), evaluate the appropriateness of different hydrological model structures across contrasting geological settings (Coxon et al., 2014) and, most recently, to diagnose deficiencies in the SEHR-ECHO GHM based on its ability to capture a range of river discharge signatures (Schaefli, 2016).

A signature-based approach within a LOA framework could also be used to compare and diagnose deficiencies in different simplified melt and runoff-routing model structures and parameterisations employed in GHMs. For this purpose, signatures need not be derived just from river discharge data, but should also be taken from other observation sources such as ice melt and snow coverage as these have shown to be useful for evaluating the consistency of GHMs across different aspects of glacio-hydrological systems (Finger et al., 2015; Hanzer et al., 2016; Mayr et al., 2013; Finger et al., 2011). By doing so, this

framework could facilitate better predictions of river flow regime changes in glaciated river basins; firstly by helping to diagnose deficiencies in GHM structures that require improvement, and secondly, by objectively selecting the range of acceptable model structures and parameterisations so that prediction uncertainty can be better constrained.

    This study is the first of its kind to apply a signature-based LOA framework for a multi-GHM-structure evaluation. The framework is used to evaluate three different melt model structures and three different runoff-routing model structures with the

aim of investigating its utility for: i) diagnosing deficiencies in the different model structures, indicating the framework's usefulness for aiding future improvement of simplified process models; and ii) constraining a prior population of model structures and parameterisations down to a smaller population of acceptable models, indicating the framework's usefulness for reducing prediction uncertainty. To do this, the models were applied to the glaciated Virkisá river basin in southern Iceland where observation data were used to derive 33 signatures of ice melt, snow coverage and river discharge against which the models were

calibrated and evaluated. LOA were defined around each signature so that acceptable and unacceptable model structures and





parameterisations could be defined. The results were first used to evaluate the capacity of the different signatures for discriminating between acceptable and unacceptable model structures and parameterisations when used individually. They were then used to compare the acceptability of the different melt and runoff-routing model structures across the range of signatures.

## 2 Methodology

### 2.1 Study site

The Virkisá river basin covers an area of 22 km$^2$ on the western side of the ice-capped Öræfajökull stratovolcano in south-east Iceland (Fig. 1). It rises from near sea level to the west, where it is bounded by steep cliffs, up to an ice-filled caldera at the summit of Öræfajökull (∼2000 m asl), the edge of which forms the basin's uppermost boundary. The basin forms a major drainage channel for accumulated ice which flows in a south-westerly direction down the steeply-sloped Öræfajökull (average slope of 0.25). It flows along two distinct glacier arms (Virkisjökull and Falljökull, hereafter referred to as Virkisjökull) around a bedrock ridge before meeting again at the terminus (∼ 150 m asl). Virkisjökull has a high mass balance gradient with a net annual accumulation of more than 7 m w.e. y$^{-1}$ at the summit (Guðmundsson, 2000) and net annual ice melt of more than 8 m w.e. yr$^{-1}$ in the main ablation zone (Flett, 2016). It has been in a phase of retreat since 1990 due to warming of the climate over this period (Hannesdóttir et al., 2015). Since 2005 the rate of retreat has accelerated to >30 m yr$^{-1}$ as a result of the detachment on the ice front from the active part of the glacier, resulting in rapid down wasting (Bradwell et al., 2013; IGS, 2017; Phillips et al., 2014). This recent rapid retreat has resulted in the formation of a small proglacial lake at the terminus which forms the headwater of the Virkisá river. The Virkisá river flows south-westerly, firstly through a 800 m bedrock-controlled section flanked on either side by push moraines from previous glacial advances. From here it continues to flow over an extensive and gently sloping sandur floodplain. The steep-sided valley walls and the relatively recent glacial maximum at the end of the Little Ice Age circa 1890 mean that there is limited soil development in and around the Virkisá river basin. Where thin soils have developed, vegetation is dominated by mosses, sparse grass and shrubs such as dwarf willow and birch.

Long-term meteorological records from two weather stations operated by the Icelandic Meteorological Office 10 km east and west of the study site show that the region experiences a maritime climate characterised by cool summers (∼ 10 $^{\circ}$C on average) and mild winters (∼ 1 $^{\circ}$C on average) with year-round precipitation (see inset in Fig. 1 bottom). The prevailing north-easterly wind and orographic lift over Öræfajökull induces a strong lateral precipitation gradient where more than two-times the precipitation falls to the east (3500 mm yr$^{-1}$) of the river basin than to the west (1500 mm yr$^{-1}$). Near-surface air temperature is mainly controlled by the altitudinal variations over Öræfajökull where the average temperature lapse rate is -0.44 $^{\circ}$C 100 m$^{-1}$ (Flett, 2016).





**Figure 1.** Location of Virkisá river basin on Öræfajökull (top) and detailed topographical map of basin including major land surface types and observation data with inset showing mean monthly climate (bottom).



## 2.2    Observation data

### 2.2.1    Climate

Several different sources of climate data are available for the study site. Measurements within the catchment are available from three automatic weather stations (AWS) installed by the British Geological Survey (BGS) between 2009 and 2011 as part of their investigation into the retreat of the Virkisjökull glacier. These are situated at 156, 444 and 805 m asl (Fig. 1) and they measure temperature, air pressure, humidity, wind speed and rainfall every 15 minutes. The lowest weather station (AWS1) is also equipped with a cosine-corrected pyranometer which measures incident shortwave radiation. Reliable, continuous time-series are available for the majority of weather variables, however, the continuity of the rainfall records are dependent on snowfall. More specifically, AWS1 is fitted with a tipping bucket rain gauge while AWS3 and AWS4 are fitted with Vaisala RAINCAP® acoustic sensors that measure the impact of individual raindrops on a steel plate. Neither of these apparatus are designed to measure snowfall. Furthermore, the presence of snowfall and/or the freezing of residual rainfall on the devices may induce erroneous measurements. Accordingly, precipitation measurements during freezing temperatures are not available.

Two additional sources of climate data are available. Firstly, the Fagurhólsmýri weather station operated by the Icelandic Meteorological Office (IMO) approximately 12 km south of the study site has daily measurements of temperature dating back to 1949 and therefore provides long-term variations in temperature around the study region. The IMO have also recently produced a gridded dataset of total precipitation as part of the ICRA atmospheric reanalysis project (Nawri et al., 2017). For this, they used the state-of-the-art HARMONIE-AROME mesoscale numerical weather prediction model (Bengtsson et al., 2017) forced by the latest ECMWF ERA-Interim reanalysis product from 1979-2016 to produce hourly precipitation data at a spatial resolution of 2.5 km over the whole of Iceland. These data provide the best estimate of long-term precipitation at the study site and, given the limited availability of precipitation measurements at higher elevations around Öræfajökull, they also provide the best estimate of spatial variations in precipitation across the region.

### 2.2.2    Ice melt

An array of 17 ablation stakes installed by Flett (2016) in the main ablation zone of the glacier between 2012 and 2014 at elevations ranging from 142 to 462 m asl provide measurements of ice melt on the glacier tongue (Fig. 1). The BGS have also undertaken annual high resolution (sub-meter) terrestrial LIDAR scans of the proglacial region including ice at the front of the glacier (see red dashed box in Fig. 1) which, given that ice flow is negligible here, provides an additional indication of ice melt.

Two digital elevation models of the ice also exist for the years 1988 and 2011 which indicate historical retreat of Virkisjökull. A 5 m 2011 DEM was constructed using high resolution airborne LIDAR scans of the ice surface (IMO, 2013) and is considered the most accurate measurement of the ice geometry and surrounding topography in the study region currently available. A 20 m 1988 DEM was derived from aerial photographs of the glacier (Landmælingar Íslands: www.lmi.is) using photogrammetric methods (Magnússon et al., 2016). Photogrammetry may suffer from errors due to image rectification and stereo-image mismatches (e.g. Barrand et al., 2009) and therefore the accuracy of this dataset is expected to be less certain, particularly over higher-elevation snow-covered terrain.





### 2.2.3 Snow coverage

Snow melt dynamics play an important role in the hydrological behaviour of mountainous catchments (Barnett et al., 2005; Jeelani et al., 2012) and as such observation data relating to the accumulation and melt of snow are important for evaluating the performance of glacio-hydrological models. No direct observations of snow accumulation or melt exist for the Virkisá river

basin and so instead, satellite snow cover data (MOD10A1 product) from the Moderate Resolution Imaging Spectroradiometer (MODIS) (Riggs and Hall, 2015) were used. These data have been archived since 2000 and consist of daily 500 m gridded maps of snow cover extent with values ranging between 0 and 1 which relate to the proportion of the ground that is snow covered. While they do not provide a direct measurement of snow mass balance, they have shown to be a useful data source for evaluating the performance of GHMs (Hanzer et al., 2016; Finger et al., 2015). The quality of the data in high latitude regions

such as Iceland are variable due to the need for good light and little or no cloud cover. As part of the MOD10A1 product, a basic estimate of the data quality is calculated as a means to avoid measurements affected by cloud cover and poor light conditions. For this study, only those data that achieved a QA score of 'good' or 'best' were used. This precluded the use of data collected between September to February presumably because of reduced daylight hours and increased cloud cover during these months.

### 2.2.4 River discharge

Hourly river discharge data collected since 2012 are available from an automatic stream gauge installed by the BGS 2 km downstream of the lake outlet on the Virkisá river (see ASG1 in Fig. 1). The gauge consists of two stilling wells with submerged pressure transducers which measure river stage and water temperature every 15 minutes. The stage data are subsequently converted into units of flow using a rating curve constructed from periodic river flow gaugings.

In conjunction with the river stage and water temperature measurements, a camera is mounted next to the river and takes photos of the channel three times a day. Given that the river is prone to freezing over the winter months, the photographic archive and temperature data were used to remove these periods from the river flow time-series. The river bed consists of large boulders (approximate diameter of 50 cm) which can become mobile during high flows causing shifts in the rating curve. For this study, river discharge data for the years 2013 and 2014 were used as gauging for these years cover a wide range of flow

magnitudes and rating shifts are limited and well constrained by observations.

### 2.3 Glacio-hydrological model

A distributed GHM which can incorporate different conceptual representations of melt and runoff-routing processes was used for all model experiments. The code was written in the object-oriented C++ programming language, making it computationally efficient and ideally suited for incorporating different model structures. The GHM consists of a 2D Cartesian grid of equally

spaced model nodes. For this study, a spatial resolution of 50 m was selected as the best balance between simulation detail and model performance. Hourly observations of precipitation, temperature and incident solar radiation were used to simulate the accumulation of snowfall and the melt of snow, firn and ice across the model domain. The snow redistribution algorithm





developed by Huss et al. (2008a) was used to account for snow drift and avalanches based on the curvature and slope of the surface. A soil infiltration and evapotranspiration model developed by Griffiths et al. (2006) solves the water balance for the non-glaciated regions of the study catchment. Excess soil moisture, rainfall and melt are then routed to the catchment outlet via a semi-distributed network of linear-reservoir cascades which represent the water storage and release characteristics of the

major hydrological pathways in the watershed. The GHM also simulates the evolution of the glacier geometry under periods of sustained negative mass balance using the Δ-h parametrisation of glacier retreat which has shown to closely reproduce the evolution of Alpine glaciers with results comparable to more complex 3-D finite-element ice flow models (Huss et al., 2010; Li et al., 2015; Van Tiel et al., 2017; Duethmann et al., 2016). Details of this and the soil water balance component of the GHM can be found in Appendix A. The following text details the different melt and runoff-routing structures adopted for this study.

### 2.3.1  Snow and ice melt model structures

Melt of snow and ice is calculated at each model node separately. Snow melt can occur at any node where a snow pack has developed. Similarly, ice melt can only occur at ice-covered nodes where the snow pack has completely melted. The mass balance at a given node is the summation of snowfall minus snow and ice melt. The GHM uses the mass balance calculated at each node to determine the equilibrium line altitude (ELA) which is updated each simulation year. A rolling three-year average

ELA was used to determine the dividing line between firn and ice on the glacier.

For this study, three different conceptual models of snow and ice melt with different levels of complexity were compared. All have been used extensively to simulate melt processes in glaciated regions around the world (e.g Gao et al., 2017; Matthews and Hodgkins, 2016; Reveillet et al., 2017; Nepal et al., 2017; Ragettli et al., 2016). The first melt model structure (TIM$_1$) employs a classic temperature index model approach (Braithwaite, 1995) whereby melt is assumed to increase linearly with

temperature above a given critical threshold:

$$M_i = \begin{cases} a_i(T - T_i^*) & T > T_i^* \\ 0 & T \leq T_i^* \end{cases} \tag{1}$$

where $a$ (m w.e. $^{\circ}$C$^{-1}$ h$^{-1}$) is the temperature factor calibration parameter that converts temperature into melt, $T$ is the near-surface air temperature and $T^*$ is the critical threshold above which melt occurs. To account for the different properties of snow, firn and ice that may bring about different values of $a$ and $T^*$, these are defined separately so that $i = (snow, firn, ice)$.

The second melt model structure (TIM$_2$) was originally proposed by Hock (1999) which includes an additional incident solar radiation term to account for topographic effects such as slope, aspect and shading which an can bring about spatio-temporal variations in melt (Arnold et al., 2006; Pellicciotti et al., 2008). Their enhanced TIM has the form:

$$M_i = \begin{cases} (T - T_i^*)(a_i + b_i \cdot SW_{\downarrow}) & T > T_i^* \\ 0 & T \leq T_i^* \end{cases} \tag{2}$$

where $b$ (m$^3$ w.e. W$^{-1}$ $^{\circ}$C$^{-1}$ h$^{-1}$) is an additional radiation factor calibration parameter that converts the measured incident solar

radiation, $SW_{\downarrow}$ (W m$^2$) into a unit melt. For this melt model structure the GHM accounts for shading using the DEM and





position of the sun in the sky which is calculated for each hourly time-step using the SPA algorithm (Reda and Andreas, 2008). Additional perturbations in solar irradiance at the surface brought about by topographic effects such as slope and aspect are accounted for by calculating the incident angle of solar radiation to scale the measured incoming radiation.

Konya et al. (2004) noted that the form of Eq. (2) is not congruent with the full energy balance equation as temperature is used to multiply the shortwave radiation term which can lead to overestimation of melt during peak temperatures. Accordingly the melt model structure proposed by Pellicciotti et al. (2005) was also used for this study (TIM$_3$) which is an enhanced TIM in additive form that also incorporates an albedo parameter, $\alpha$:

$$M_i = \begin{cases} a_i(T - T_i^*) + b_i \cdot SW_\downarrow (1 - \alpha_i) & T > T_i^* \\ 0 & T \leq T_i^* \end{cases} \tag{3}$$

where $b$ has the units m$^3$ w.e. W$^{-1}$ h$^{-1}$. Following Pellicciotti et al. (2005), this melt model structure also includes the dynamic snow albedo algorithm proposed by Brock et al. (2000) which accounts for the drop in snow albedo as it ages using a logarithmic function with the form:

$$\alpha_{snow} = p_1 - p_2 \cdot log_{10} \cdot T_a \tag{4}$$

where $p_1$ is the albedo of fresh snow (set to 0.9), $p_2$ is an empirical calibration parameter and $T_a$ is the accumulated daily maximum temperature greater than 0$^o$C since snowfall.

For all melt model structures in the GHM, melt $M$ is converted into a volumetric melt $M_v$ at each node:

$$M_v = M \cdot A \tag{5}$$

where $A$ is the model node area. Following Hopkinson et al. (2010) the area of each node is corrected for surface slope:

$$A = \frac{L^2}{\cos \beta} \tag{6}$$

where $L$ is the model node length and $\beta$ is the node surface slope.

### 2.3.2 Runoff-routing model structures

Runoff includes any rainfall falling on, and melting of the snow and ice as well as excess soil moisture from those areas free of ice and snow. The concept of linear reservoirs was employed to route this runoff to the catchment outlet. A linear reservoir receives a volumetric inflow and releases it at a rate proportional to its internal water storage following:

$$q = \frac{1}{k} s \tag{7}$$

where $q$ is the outflow (m$^3$ h$^{-1}$), $s$ is the storage (m$^3$) and $k$ is mean residence time of the reservoir (h) which accounts for the diffusive effect of storage and release mechanisms within the catchment. Increasing the value of $k$ increases the diffusion effect on the inflow hydrograph. Additional controls on the diffusion and lag effects can be obtained by arranging a cascade of



multiple linear reservoirs in series (Ponce, 2014) so that the outflow from the previous reservoir is the inflow for the subsequent reservoir. With this setup, the continuity equation for the $jth$ reservoir of $n$ reservoirs in series, where $j = (1, 2...n)$ can be written as:

$$\frac{ds_j}{dt} = \begin{cases} i - q_j & j = 1 \\ q_{j-1} - q_j & j > 1 \end{cases} \tag{8}$$

The outflow hydrograph is then taken from $q_n$.

For temperate glaciers, the common practice is to subdivide the catchment into one or more hydrological response units (HRU) which are thought to have different water storage and release characteristics. For example, the firn, snow and bare ice have generally shown to respond over relatively long, intermediate and short time-scales respectively (Hock and Jansson, 2005) and therefore these may be characterised as separate HRUs, although as noted previously, simpler and more complex

definitions of HRUs have been defined in the past. Subsequently, three runoff-routing model structures were proposed with different levels of complexity structured around these subdivisions (Fig. 2).

The first and simplest runoff-routing model structure (ROR$_1$) uses a single linear reservoir cascade (e.g. Boscarello et al., 2014) to route the inflow from all runoff sources simultaneously. This structure makes no distinction between the different runoff sources and flow pathways and assumes that all conform to the same storage-discharge relationship.

The second model structure (ROR$_2$), employs two linear reservoir cascades in parallel (e.g. Hannah and Gurnell, 2001). The first cascade represents the slow percolation of water through the snow and firn HRUs, while the second cascade represents faster flow of water through the bare ice and overland. This approach therefore makes some distinction between the different flow pathways and, by conditioning the parameters so that the snow and firn have a more diffuse response function, it introduces a degree of non-linearity in the discharge response to runoff.

The third runoff-routing model structure (ROR$_3$) has not been used previously. It employs separate linear reservoir cascades to route water from the firn, snow, ice and soil HRUs. Here the parameters are conditioned so that the firn is the most diffuse, slowly responding reservoir, followed by the snow and then the ice and soil zones are considered to be relatively flashy, quickly responding HRUs. This approach also includes some representation of linkages between these various units. Here it is hypothesised that water that flows through the firn, must then flow through the downstream bare ice HRU before it reaches the

river. Similarly, water that percolates through the snow pack must also flow via the HRU that it overlies before it reaches the river. There are therefore six different flow pathways that runoff may take before reaching the river outlet (see Fig. 2c) and this represents the most complex, non-linear runoff-routing model structure.

## 2.4 Driving climate data

The GHM was configured to run from the initial ice geometry of 1988 through to 2015. It requires continuous measurements

of hourly precipitation, near-surface air temperature and incident solar radiation to drive the various model components.





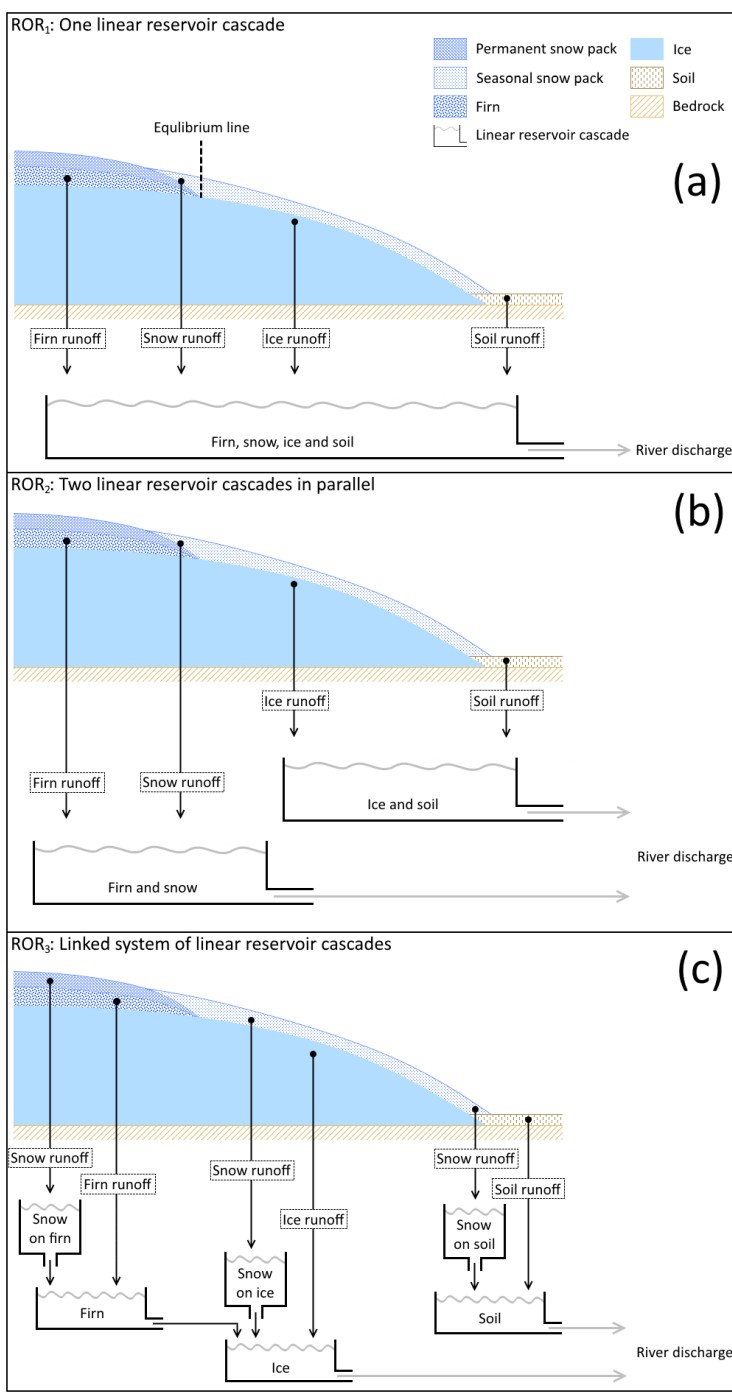

**Figure 2.** Three runoff-routing model structures which relate the linear reservoir cascade configurations to idealised cross-sections of a temperate glacier.





### 2.4.1 Precipitation

A new gridded precipitation time-series was constructed for the GHM that incorporates the measurements of rainfall from the weather stations in the Virkisá basin and the information on spatial and long-term variations in precipitation from the gridded ICRA reanalysis product. First, the weather station rainfall data were used to bias-correct the ICRA reanalysis product. Given that none of the weather stations are equipped with devices to measure snowfall, and that freezing temperatures can induce erroneous measurements in rainfall, only data with three consecutive preceding above-freezing days were used. This is a major issue for using AWS4 as the majority of days, particularly in the winter, are below freezing at this elevation. Accordingly, the AWS4 rainfall data were not used for the bias correction procedure. Furthermore, because the AWS1 and AWS3 gauges overlap the same ICRA data pixel, and because the AWS1 time-series is the longest and most complete, it was decided to use the AWS1 data to bias-correct the overlapping ICRA data pixel. Here, the equidistant quantile mapping (EQM) approach (Li et al., 2010; Srivastav et al., 2014; Sachindra et al., 2014) was employed to bias-correct the ICRA precipitation time-series. EQM is an adaptation of the original quantile mapping method that accounts for non-stationarity in the moments of the biased time-series and helps to preserve changes in the cumulative distribution function of the precipitation data that may have occurred over time (Switanek et al., 2017; Cannon et al., 2015). To evaluate the effectiveness of the bias-correction procedure, the $R^2$ correlation score was calculated between the bias-corrected ICRA data and the measured AWS1 data. At an hourly timescale, the ICRA data only captured 22% of the observed variance in the AWS1 rainfall record. However, when averaged to a daily timescale the $R^2$ score increased to 0.49, and for a three-daily average timescale the $R^2$ increased to 0.72. The limited accuracy of the ICRA precipitation data at an hourly timescale could hinder the acceptability of the GHM across some of the signatures (e.g. the river discharge signatures related to the timing of flows). However, the AWS1 rainfall record is complete for the years 2013 and 2014 where the GHM is compared against observed river discharge signatures. As such, poor replication of the timing of hourly rainfall events should not influence the GHM's ability to capture the river discharge signatures. Rather, the role of the bias-corrected ICRA precipitation data was primarily to drive the glacier-mass balance component of the GHM prior to 2009 for which reliable precipitation data on a three-daily timescale were deemed adequate.

### 2.4.2 Near-surface air temperature

The longest record of hourly temperature measurements in the Virkisá river basin are from AWS1 which starts in 2009. To generate a continuous time-series of temperature back to 1988, daily measurements of temperature available from the nearby Fagurhólsmýri weather station were used. A comparison of daily average temperatures showed there to be a good linear relationship between the two stations with an $R^2$ of 0.92. As such, this linear model was used to bias-correct the daily weather station data so that it could be combined with the AWS1 time-series. To downscale the data to an hourly resolution, 24-hour temperature anomalies were randomly sampled from the AWS1 record, thereby ensuring the complete time-series had a consistent sub-daily variability. Of course, diurnal cycles in temperature are dependent on the time of year, whereby increased incident solar radiation in the summer enhances sub-daily temperature variability. Therefore, the sampling strategy was employed on a month-by-month basis. The complete hourly time-series of temperature at AWS1 is shown in Fig. 3b.



**Figure 3.** Continuous hourly time-series of precipitation (a), temperature (b) and incident solar radiation (c) between 1988 and 2015 at AWS1.



As in many glaciated catchments topography, to a large extent, controls spatial temperature variations. The importance of characterising temperature lapse rates for glacio-hydrological modelling is well known because it has a strong control on spatial patterns of melt simulations (Gardner and Sharp, 2009; Heynen et al., 2013; MacDougall et al., 2011). In fact while many studies employ a fixed temperature lapse rate, in reality seasonal variations in surface characteristics (e.g. albedo and

roughness) and atmospheric conditions can bring about strong seasonal and diurnal variations in lapse rates which control melt processes (Gardner et al., 2009; Minder et al., 2010; Immerzeel et al., 2014). Furthermore, local atmospheric phenomena associated with mid-latitude glaciers such as katabatic winds which bring cool dense air over the ice surface can serve to shallow the temperature gradient (Petersen and Pellicciotti, 2011; Ragettli et al., 2014). Having analysed near-surface air temperature variations both on and away from the Virkisjökull glacier, it was deemed most appropriate to extrapolate temperature across

the study catchment using a seasonally variable hourly lapse-rate in conjunction with an on-ice temperature correction function based on the work of Shea and Moore (2010) (see Appendix B).

### 2.4.3 Incident solar radiation

The only source of incident solar radiation is the continuous hourly time-series from AWS1. To construct a continuous time-series back to 1988, a resampling strategy was employed to generate a complete time-series that was statistically consistent with

the data at AWS1. It was found that during the summer months, the daily range in incident solar radiation and temperature are strongly correlated. Therefore, when generating a continuous time-series of hourly incident solar radiation from 1988, it was important to maintain this dependence between intra-day solar radiation and temperature variability. To do this, a coordinated (in time) sampling strategy identical to that used for the near surface air temperature data was employed. More specifically, for each random 24-hour temperature anomaly sample from the AWS1 record used to build part of the temperature time-series, the

corresponding 24-hour solar cycle data were extracted and used to build the same part of the incident solar radiation time-series. Figure 3c shows the complete time-series of incident solar radiation used to drive the model.

### 2.5 Signatures and limits of acceptability

Observations of ice melt, snow coverage and river discharge were used to derive 33 unique signatures with LOA to characterise the glacio-hydrological behaviour of the Virkisá river basin over different spatio-temporal scales and evaluate the acceptability

of the different model structures (Table 1). For convenience, the signatures have also been subdivided into 11 attributes which encapsulate the main aspects of model behaviour that were assessed.

### 2.5.1 Ice melt

The average winter (November 2012 - April 2013) and summer (May 2013 - September 2013) melt across the ablation stake network were used to characterise the short-term, seasonal ice melt on the glacier tongue. Of course, point measurements of

melt are not directly comparable to simulated melt at the GHM nodes as these simulations represent the average melt over the node area. Therefore, the GHM can only be expected to get as close to the stake measurements as the actual spread in melt over





**Table 1.** Summary of signatures used to evaluate model acceptability

| Group | Attribute | Attribute ID | Signature | Limits of acceptability |
|---|---|---|---|---|
| Ice melt | Seasonal ice melt on tongue | Seas melt | 2013 Summer ice melt | 5.22 – 6.44 m we |
| | | | 2012-2013 Winter ice melt | 0.64 – 0.78 m we |
| | Long term glacier volume change | Melt vol | Change in ice volume (1988-2011) | -0.36 – -0.28 km$^3$ |
| Snow coverage | Snow coverage in lower catchment | Low snow | Mean snow coverage in spring | 0.32 – 0.45 |
| | | | Mean snow coverage in early summer | 0.02 – 0.08 |
| | | | Mean snow coverage in late summer | 0.00 – 0.03 |
| | Snow coverage in mid catchment | Mid snow | Mean snow coverage in spring | 0.70 – 0.80 |
| | | | Mean snow coverage in early summer | 0.17 – 0.27 |
| | | | Mean snow coverage in late summer | 0.00 – 0.04 |
| | Snow coverage in upper catchment | Upp snow | Mean snow coverage in spring | 0.81 – 0.90 |
| | | | Mean snow coverage in early summer | 0.51 – 0.64 |
| | | | Mean snow coverage in late summer | 0.02 – 0.09 |
| River discharge | Mean monthly river flow | Mnthly flow | Mean January river flow | 1.16 – 1.86 m$^3$ s$^{-1}$ |
| | | | Mean February river flow | 1.69 – 2.92 m$^3$ s$^{-1}$ |
| | | | Mean March river flow | 0.85 – 1.58 m$^3$ s$^{-1}$ |
| | | | Mean April river flow | 0.73 – 1.48 m$^3$ s$^{-1}$ |
| | | | Mean May river flow | 1.50 – 2.16 m$^3$ s$^{-1}$ |
| | | | Mean June river flow | 4.12 – 6.23 m$^3$ s$^{-1}$ |
| | | | Mean July river flow | 6.33 – 10.30 m$^3$ s$^{-1}$ |
| | | | Mean August river flow | 5.72 – 9.15 m$^3$ s$^{-1}$ |
| | | | Mean September river flow | 4.55 – 7.38 m$^3$ s$^{-1}$ |
| | | | Mean October river flow | 3.88 – 7.02 m$^3$ s$^{-1}$ |
| | | | Mean November river flow | 3.90 – 7.40 m$^3$ s$^{-1}$ |
| | Quick release high flows | High flows | Volume under highest flow section of FDC | 59.4 – 116.0 |
| | | | Slope of highest flow section of FDC | 2.67 – 9.88 |
| | | | Volume under high flow section of FDC | 70.6 – 111.0 |
| | | | Slope of high flow section of FDC | 0.38 – 0.79 |
| | Slow release low flows | Low flows | Volume under low flow section of FDC | 20.9 – 46.1 |
| | | | Slope of low flow section of FDC | 0.03 – 0.05 |
| | Flow variability | Flow var | Coefficient of variation | 0.95 – 1.83 |
| | Melt runoff timing | Melt timng | Peak summer flow hour | 17:00 – 18:00 |
| | Flashiness | Flow flash | Integral scale | 25 – 44 h |
| | | | Rising limb density | 0.13 – 0.20 |




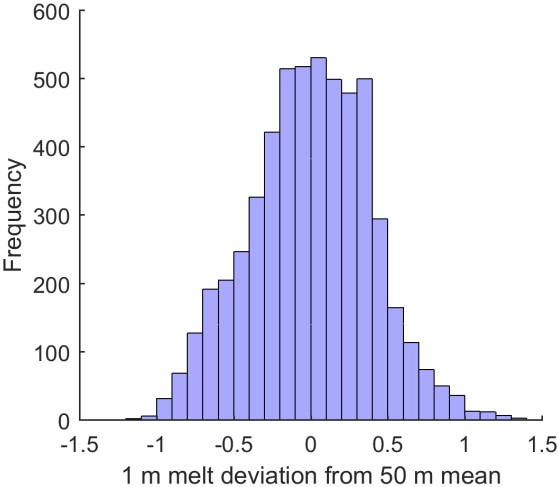

**Figure 4.** Histogram of deviation of 1 m melt from 50 m mean derived from terrestrial LIDAR scans of static ice front between 2012-2014.

the equivalent model node area. To calculate this spread, the high resolution terrestrial LIDAR scans taken during the ablation stake campaign were used. The scans were used to estimate the spread of melt deviations from the mean melt across 50 m square regions (Fig. 4). The 95% confidence bounds ($\pm$ 0.78 m yr$^{-1}$) were then used to define the LOA around the winter and summer melt signatures where it was assumed that the spread should be proportional to the total melt. This assumption leads to much narrower LOA around the winter melt signature than the summer melt signature.

A signature to characterise the long-term change in glacier volume was also quantified by differencing two 3D models of the ice from 1988 and 2011. These models were constructed using the two ice surface DEMs in combination with a bedrock model of the Öræfajökull region (Magnússon et al., 2012). Given the potential errors in the 1988 DEM, this dataset was assumed to be the main source of uncertainty in the calculation of the ice volume change signature. A comparison to the more accurate 2011 DEM shows that the 1988 DEM captures the gridded elevation data across the non-glaciated portion of the study area with reasonable accuracy (Fig. 5a). The residuals are approximately normally distributed with a mean error of zero (Fig. 5b) and they show to be largest for those parts of the catchment that are steeply sloped (scatter in Fig. 5c). To account for these errors in the calculation of the ice volume change signature, 1000 unique DEMs of the 1988 ice surface were generated by randomly perturbing each pixel of the original dataset with perturbations drawn from a normal distribution with mean zero. Given that the spread of the residuals increases for those areas of the catchment that are steepest, the shape parameter of the error distribution (standard deviation) was varied according to the slope of each pixel of the 1988 DEM (see dark blue line in Fig. 5c). From these, 1000 equally probable estimates of ice volume change were calculated and the 95% confidence interval was used to define the LOA. The total change in ice volume over 23 years from 1988 was estimated to be between -0.36 and -0.28 km$^3$.

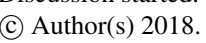



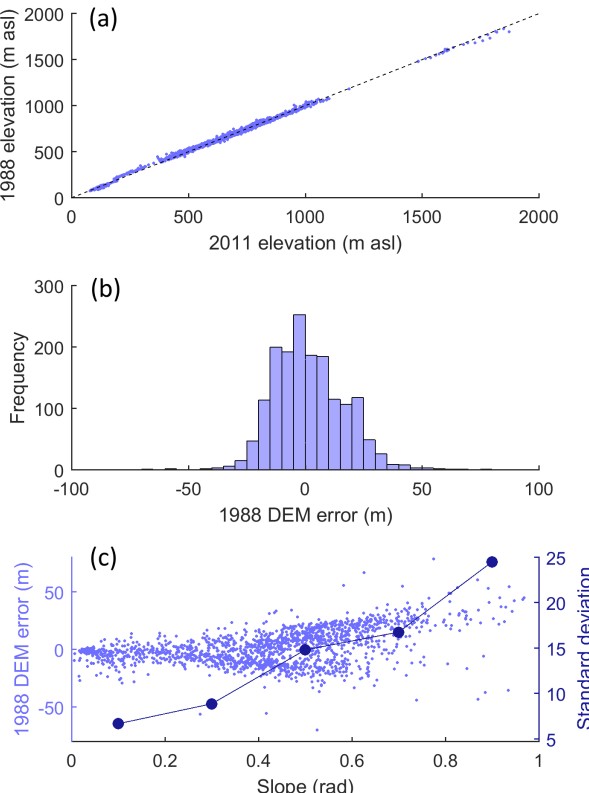

**Figure 5.** Error model for estimating uncertainty in glacier volume change between 1988 to 2011 including: 1988 vs 2011 off-ice DEM elevations (a), distribution of 1988 DEM errors calculated as difference between 1988 and 2011 off-ice elevations (b) and estimation of change in standard deviation of errors with DEM slope (c).

### 2.5.2 Snow coverage

Having removed the MODIS data that did not pass the QA test including all of the data between September and February, less than 5% of the remaining data were usable, and therefore, it was decided that these data should be combined to derive three seasonal average snow coverage maps. From these maps, three snow coverage curves were constructed that define the mean

5   catchment snow coverage over an elevation range for three different times of the year: spring (March and April), early summer (May and June) and late summer (July and August) (Fig. 6). The curves, provide information on both the spatial and temporal distribution of snowfall in the study catchment. They were constructed by distributing the seasonal average snow distribution maps across the 50 m model grid DEM. For example, for a MODIS pixel value of 0.5, 50 of the corresponding DEM pixels were assumed to be snow covered. The MOD10A1 product cannot distinguish between snow and ice-covered regions, so only

10   data that covered ice-free parts of the catchment were used. This limited the analysis up to a maximum elevation of just under 1200 m asl. While this does not cover the full elevation range of the catchment, Fig. 6 shows that the three curves capture a





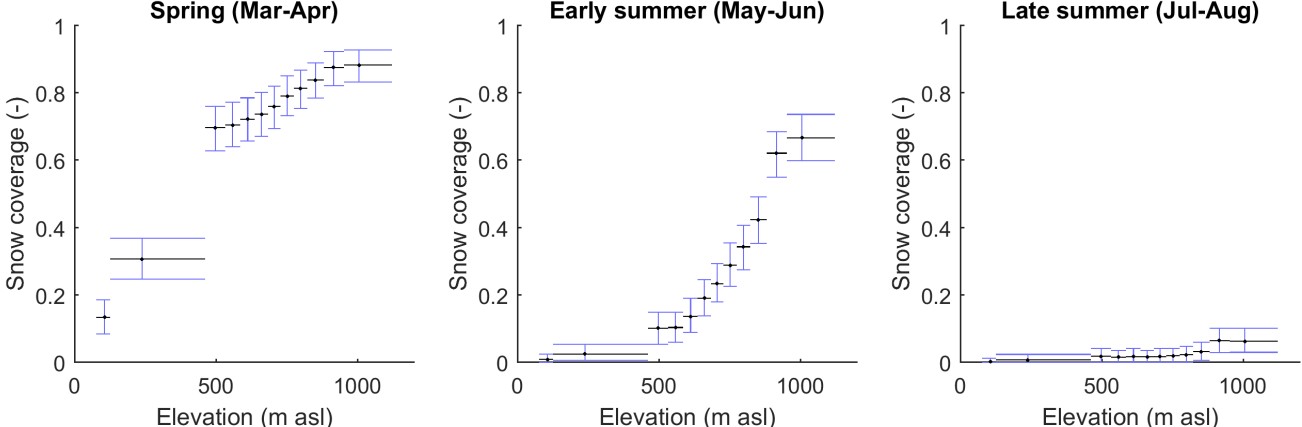

**Figure 6.** Snow coverage curves defined from the MOD10A1 snow cover product from 2000 - 2015 with 95% confidence bounds.

large amount of variability in seasonal snow cover. From the three snow coverage curves, the mean snow coverage from the lower, middle, and upper terciles of the curves were used as signatures of snow coverage.

There exists no definitive quantification of errors in the MOD10A1 product that can be used to estimate LOA for these signatures. Previous validation of the MODIS data using satellite imagery has shown the data to be relatively robust (Salomonson
5  and Appel, 2004). Accordingly, it was assumed that as with the ablation stake data, the primary source of uncertainty stems from scale differences between the data and the model simulations. More specifically, because the MODIS data have a coarser resolution (500 m) than the DEM over which the MODIS data were distributed (50 m), a MODIS pixel value of 0.5 only indicates that 50 of the corresponding 100 DEM pixels are snow covered. The construction of a snow distribution curve, therefore necessitates some assumptions about where the snow actually lies which will influence the shape of the snow distribution
10  curve. Accordingly, the LOA were quantified to account for this uncertainty. Here, for each of the seasons, a mean MODIS snow cover map over the study region was derived. Then, for each 500 m pixel, snow was randomly distributed across the corresponding DEM pixels 1000 times. From these, an equal number of snow distribution curves and corresponding snow distribution signatures could be derived, each assumed to be equally probable. The 95% confidence bounds from this distribution of snow cover signatures were used to define the LOA which are indicated by blue error bars in Fig. 6.

15  ### 2.5.3  River discharge

The hourly river discharge data for the years 2013 and 2014 measured at ASG1 (Fig. 7a) were used to define 21 different river discharge signatures that cover a range of temporal scales and flow magnitudes. The majority of these signatures were based on previous studies (Coxon et al., 2014; Yilmaz et al., 2008; Westerberg et al., 2016; Shafii and Tolson, 2015; Hrachowitz et al., 2014; Schaefli, 2016; Viglione et al., 2013; Euser et al., 2013; Garavaglia et al., 2017; Yadav et al., 2007; Casper et al., 2012;



**Figure 7.** River flow time-series from ASG1 with quantified confidence intervals (a), rating curve uncertainty used to quantify confidence intervals (b) and zoomed section of river flow time-series (see yellow dash box in top plot) with confidence intervals (c).

Clausen and Biggs, 2000; Teutschbein et al., 2015; Andrés-Doménech et al., 2015; Monk et al., 2007; Sawicz et al., 2014; Winsemius et al., 2009) and are summarised as follows.

Mean monthly river flows were calculated to characterise the seasonal river flow regime. Signatures were also derived from sections of the flow duration curve to characterise quick-release high flows and slow-release low flows. These include signatures that quantify the volume under the section (flow magnitude) and the slope of section (flow variability) for the low flow section





(99-66% flow exceedance), high flow section (15-5% flow exceedance) and highest flow section (5-0.5% flow exceedance). An overall estimate of flow variability, the coefficient of variation, was also calculated. Related to this, two further signatures, the rising limb density and integral scale, provide a measure of flashiness. The rising limb density is the ratio of number of flow peaks to the total time to peak where a higher number is more flashy. The integral scale measures the lag time at which

the autocorrelation function of the flow time-series falls below $\frac{1}{e}$ (diurnal cycles in river flow were removed prior to this using a moving average filter). A higher integral scale therefore indicates a more slowly responding hydrological system. Finally, the peak summer flow hour of the observed discharge time-series was calculated to characterise the intra-day river discharge response to melt.

Estimates of river discharge are inherently uncertain (Pappenberger et al., 2006). McMillan and Westerberg (2015) provide

a useful definition of two important sources of uncertainty which they distinguish as either aleatory (random) or epistemic (of an unknown character). The first stem from random measurement errors such as those from the instrument used for periodic river gaugings. These cause gauging points to vary around the 'true rating curve', typically according to some formal statistical definition. Epistemic uncertainty stems from the assumptions hydrologists have to make when constructing rating curves such as assuming the river bed profile and horizontal flow velocity distribution is relatively stable over time. These errors make

fitting a single rating curve to all of the gauging data invalid. Accordingly, McMillan and Westerberg (2015) propose a method to define rating curve uncertainty which accounts for both sources of error and has been used to estimate uncertainty in river discharge signatures (Westerberg et al., 2016). The random error component was defined from analysis of 27 flow gauging stations in the UK with stable ratings and without obvious epistemic errors (Coxon et al., 2015). They conclude that this source of error is best approximated by a logistical distribution model. To account for epistemic error, they reject the assumption that

the rating curve is fixed in time and instead they fit an ensemble of rating curves to all of the gauging data. Each curve is weighted by a 'Voting Point' likelihood function which scores it based on how many points of the periodic gaugings it is able to intersect (and at what location in the logistical distribution of each measurement).

In this study, the methodology proposed by McMillan and Westerberg (2015) was used to estimate rating curve uncertainty. Markov chain Monte Carlo sampling was used to define 667 unique rating curves which together define the rating curve

uncertainty (Fig. 7b). From these an equivalent distribution of each river discharge signature was derived from the ensemble of flow time-series (Fig. 7c), from which the 95% confidence bounds were used as the LOA. Because the Voting Point method only accounts for uncertainty in the flow magnitude and not the timing, it was not suitable to apply this approach to the three signatures that characterise melt runoff timing and flashiness. For these signatures, Schaefli (2016) proposed that the LOA should be derived by subsampling different periods of the flow time-series. For this study a month-by-month sub-sampling

strategy was employed to do this.

## 2.6 Model calibration procedure

The GHM was configured to run from 1988 to 2015 so that simulations could be compared against all observation signatures. The initial ice surface was set to the 1988 DEM of the ice while the bedrock and land surface topography were taken from the Öræfajökull bedrock map (Magnússon et al., 2012). Initial snow coverage, soil moisture, linear reservoir storages and ELA




were determined by running the model for three consecutive years prior to the simulation period using climate data from 1985 to 1988.

In total there were nine possible structural configurations of the GHM including all possible combinations of the three melt and runoff-routing model structures. For each of the nine configurations, the melt and runoff-routing model parameters were

calibrated to achieve the closest fit to the observed signatures. To do this, first a set of preliminary runs were undertaken to assess the sensitivity of the simulations to the parameters. Here, it was found that the simulations were insensitive to the firn melt parameters across the range of 33 signatures. Accordingly, these were set to the same values as for snow. Similarly, none of the signatures were sensitive to the threshold above which melt occurs, $T^*$, and accordingly, this was set to 0 °C throughout the model experiments. Finally, it was also decided to fix the albedo parameter for ice in TIM$_3$ to 0.3. This was because this

parameter directly interacts with the $b$ parameter and therefore provides no extra control over model behaviour.

The remainder of parameters were kept for calibration (see Table C1). For each GHM configuration, 5000 Monte Carlo simulations with random parameter sets sampled from pre-defined uniform distributions were undertaken. The prior parameter distributions were defined from a review of previous modelling studies and later refined during the preliminary runs noted above. The quasi-random Sobol sampling strategy (Brately and Fox, 1988) was employed to sample the parameter space as

efficiently as possible. The simulated signatures from each model run (parameter set) were then evaluated against the observed signatures using a continuous acceptability score:

$$s_j = \begin{cases} 0 & low_j \leq sim_j \leq upp_j \\ \frac{sim_j - upp_j}{upp_j - obs_j} & sim_j > upp_j \\ \frac{sim_j - low_j}{obs_j - low_j} & sim_j < low_j \end{cases} \tag{9}$$

where $obs_j$ and $sim_j$ are the observed and simulated values for signature $j$ and $upp_j$ and $low_j$ are the upper and lower LOA. Here, a score of zero indicates that the model captures the observed signature within the LOA. An absolute score greater than

0 is outside of the LOA and therefore unacceptable. The sign of the score indicates the direction of bias while its magnitude indicates the model's performance relative to the LOA. A score of -3 would indicate that the model underestimates the signature by three times the observation uncertainty.

Given that there are 33 different signatures to calibrate to simultaneously, it was important to define a weighting scheme to achieve the best overall performance across the range of signatures. It was decided that, for a given GHM configuration, the

5000 runs should be ranked by a weighted average score where each group, each attribute within each group and each signature within each attribute were given equal weighting so that the scores were not biased to a particular group or attribute. The top 1% of model runs that achieved the smallest weighted average acceptability scores were then taken as the calibrated models for each GHM configuration and the average acceptability scores of these are reported. A bootstrapping with replacement re-sampling scheme was also used to assign 95% confidence intervals around all reported acceptability scores. While not a

formal test of statistical significance, these were used to avoid reporting differences between the GHM configurations where issues such as under-sampling of the parameter space would make such conclusions unjustified. Where confidence intervals do not overlap, differences are hereafter referred to as substantial. The different GHM configurations were also compared when





calibrated to individual groups of signatures (ice melt, snow coverage and river discharge). In this case the same weighting procedure was applied to a single group only.

## 3 Results

### 3.1 Signature discrimination power

As a first step towards evaluating the LOA framework, the discrimination power of the signatures was investigated to determine their relative usefulness for discriminating between acceptable and unacceptable model structures and parameterisations when used individually. A total of 45,000 calibration runs, each with unique model structures and parameterisations (hereafter referred to as model compositions) were undertaken in this study. The signatures with the highest discrimination power were defined as those that best constrain the range of acceptable model compositions. Here, the total number of acceptable model

compositions were calculated for each signature as an indicator of discrimination power (bars in Fig. 8a). The results indicate that the ice melt signatures are the best discriminators, where each accepted less than 5000 model compositions. Of these, the winter melt signature from the ablation stake measurements is the best discriminator while the summer melt signature shows the least discrimination power. The snow coverage signatures generally show to be inferior discriminators when compared to the ice melt signatures. The late summer snow coverage signature for the lower catchment shows to be the poorest discrimina-

tor, presumably because there is negligible snow cover here at this time of the year; an observation that almost all of the model compositions have no difficulty in replicating. In contrast, no model compositions are deemed acceptable for the signatures of the spring and early summer snow coverage in the upper catchment. This could indicate a deficiency in the melt model structures tested in this study and is considered in the remainder of the analysis.

The discrimination power of the river discharge signatures shows to be highly variable, but there are several discernible

patterns. Firstly, the mean monthly flow signatures between January and June, when river discharge is low, show to be better discriminators than the higher-flow signatures from July to October. The mean monthly January and May flows stand out as being particularly powerful at discriminating between acceptable and unacceptable model compositions suggesting that these are likely to be important focal points for characterising model deficiencies. Those signatures related to the variability of flows such as the coefficient of variation and the flow duration curve slope signatures, as well as peak flow hour (timing) and rising

limb density (flashiness) also show to be relatively good discriminators.

To determine the structural discrimination power of each signature, the total number of GHM configurations that returned at least one acceptable simulation has also been calculated for each signature (scatter in Fig. 8a). They show that when used individually, most of the discrimination power stems from constraining the parameter space rather than constraining the structural space. Only the lower-catchment spring snow coverage and mean January river flow signatures discriminate between structures

where only six of the nine GHM configurations returned acceptable simulations. In both cases it was the GHM configurations that employed the TIM$_3$ melt model structure that could not capture these signatures within their LOA.

To indicate of how each signature helps to reduce river flow prediction uncertainty, a second measure of discrimination power has also been calculated (Fig. 8b). Here, the mean simulated range in river discharge from the population of acceptable models





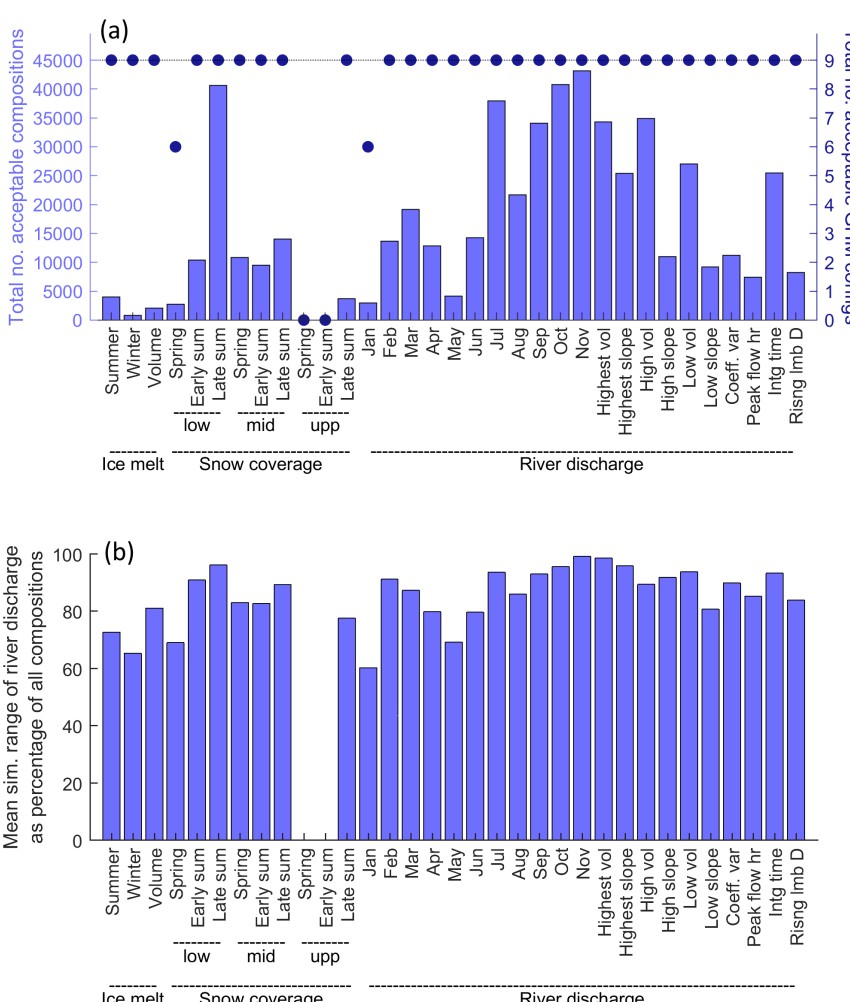

**Figure 8.** Total number of acceptable model compositions and configurations for each signature (a) and mean simulated range in river discharge from the population of acceptable models as a percentage of the simulated range using all of the 45000 model compositions (b).





has been calculated as a percentage of the simulated range using all of the 45,000 model compositions for each signature. These results show that when used individually, all of the signatures help to constrain the river flow prediction uncertainty, although the effectiveness of each is variable. The mean January and May river flow signatures again exhibit good discrimination power, reducing the mean river discharge uncertainty to 60-70% of that from the full population of model compositions.

Similarly, the winter ice melt and spring snow coverage in the lower catchment remain as two of the best discriminators. However, some signatures such as the long-term volumetric change in the glacier, which showed to be a good discriminator of model acceptability, are not as effective at reducing river discharge prediction uncertainty. Overall, these results highlight the contrasting discriminatory power of the different signatures employed in this study.

## 3.2   Acceptability of melt model structures

While all signatures clearly demonstrate discrimination power when used individually, it remains to be seen how effective the LOA framework is for discriminating between and diagnosing deficiencies in different model structures when using multiple evaluation criteria. Here, the acceptability scores obtained after calibrating the GHM to the different groups of signatures (ice melt, snow coverage and river discharge) using the three different melt model structures have been calculated (Fig. 9). The light grey boxes indicate those signatures that have been captured within the LOA and the dark grey boxes and their corresponding

acceptability scores indicate those signatures for which the structures were not able to capture within the LOA. So that the river discharge acceptability scores can be compared fairly, they have all been obtained using the $ROR_1$ runoff-routing structure.

When calibrated against the ice melt signatures, the GHM is not able to capture them within their LOA, regardless of the melt model structure used. The different GHM configurations show a tendency to overestimate the measured summer and winter melt from the ablation stake data, yet underestimate the long-term change in total ice volume (note underestimation

here refers to the simulated loss in ice volume). This highlights a deficiency in the melt model structures as they are unable to reconcile the three melt signatures simultaneously within the observation uncertainty. The winter melt is by far the most unacceptable simulation, particularly when using the $TIM_1$ structure where it is overestimated by more than 30 times the observation uncertainty.

Each of the GHM configurations using the three melt model structures have been ranked from 1 to 3 in the top left corner of

each box where the acceptability scores are substantially different (Fig. 9). While there are clearly differences in the acceptability scores for the summer melt and ice volume signatures, these are not substantially different and therefore it is not possible to say that one structure is more acceptable than another. Indeed, a comparison of the simulated ice thickness change along the Falljökull and Virkisjökull arms of the glacier reveal that all three configurations of the GHM produce almost identical simulations which broadly capture the observed ice thickness change between 1988 to 2011 (Fig. 10).

For the winter melt signature, there is a substantial difference in acceptability when using the three melt model structures. Here, the GHM configuration using the $TIM_3$ structure is the most acceptable while that using the $TIM_1$ structure is least acceptable, indicating that while all configurations produce simulations outside of the LOA, there is an improvement in ice melt simulations when implementing the most sophisticated $TIM_3$ melt model structure.





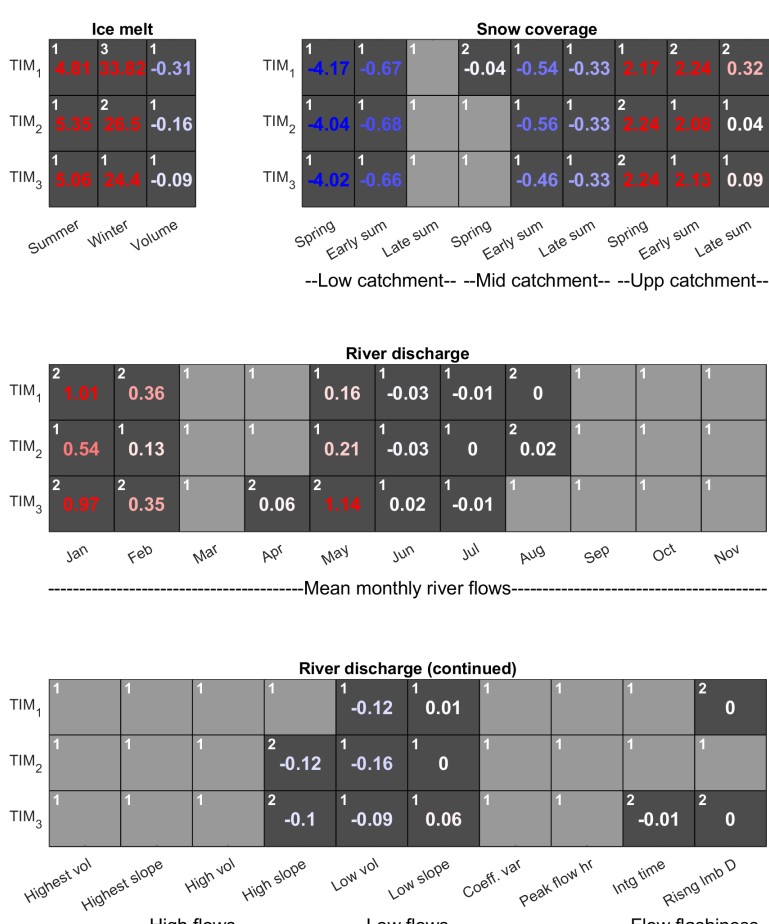

**Figure 9.** Acceptability scores obtained after calibrating the GHM using the three melt model structures in combination with the ROR$_1$ runoff-routing model structure. The three GHM configurations were calibrated against ice melt, snow coverage and river discharge signatures separately. Light grey boxes indicate acceptable simulations ($s = 0$) and numbered, dark-grey boxes indicate unacceptable simulations coloured blue and red to indicate negative and positive bias respectively. White numbers in top left of each box indicate relative ranking where acceptability scores are substantially different between the GHM configurations.





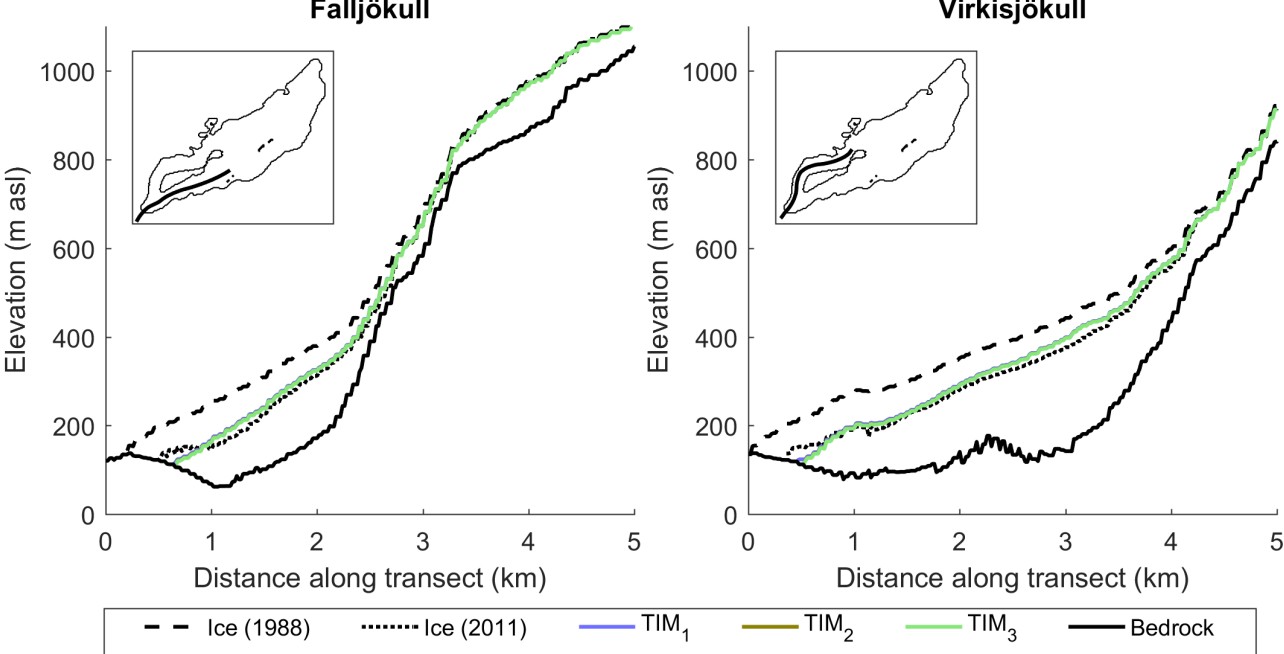

**Figure 10.** Observed and simulated ice thickness change as measured along transects of the Falljökull and Virkisjökull glacier arms. Insets show transect location.

For the snow coverage signatures, all three of the GHM configurations capture the late summer snow coverage in the lower portion of the catchment within the LOA. When using the $TIM_2$ and $TIM_3$ structures the mid-catchment spring snow coverage is also captured. The remaining snow coverage signatures are not captured within the LOA where all configurations show a tendency to underestimate snow coverage in the lower and mid parts of the catchment and overestimate snow coverage in the upper part of the catchment. To investigate why this is, Fig. 11 (left) shows the simulated early summer mid-catchment and upper-catchment snow coverage signatures for the 5000 calibration parameter sets (blue dots) used with the $TIM_1$-$ROR_1$ GHM configuration. Here it can be seen that regardless of the choice of melt model parameters, this structure is not able to capture both of these signatures within their LOA simultaneously (indicated by yellow area). A similar inconsistency exists when comparing snow coverage over different seasons where the GHM is not able to capture the lower catchment snow coverage in the early summer and spring simultaneously (Fig. 11, right). Indeed, this inconsistency extends across all melt model structures.

A comparison of simulated snow distribution curves from the calibrated models (Fig. 12) reveals that all return similar simulations. The simulation using $TIM_1$ deviates slightly from the curve produced by the GHM when using the $TIM_2$ and $TIM_3$ structures, but overall the choice of melt model structure has a limited influence on the simulated seasonal snow coverage.

The acceptability scores for the river discharge signatures in Fig. 9 show that regardless of the choice of melt model structure, when used in conjunction with the $ROR_1$ runoff-routing model structure, all are able to capture a range of the river discharge



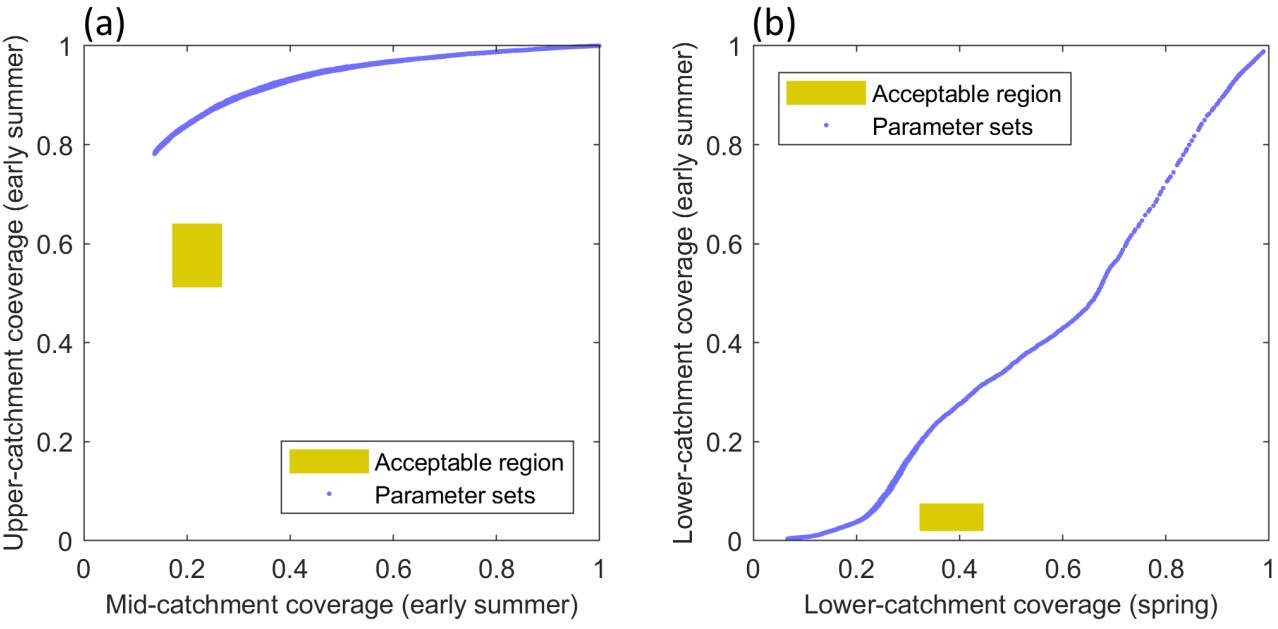

**Figure 11.** Simulated snow coverage signatures from the 5000 calibration runs (blue dots) for the TIM$_1$-ROR$_1$ GHM configuration including: early summer mid-catchment and upper-catchment snow coverage signatures (left), and lower-catchment spring and early-summer snow coverage signatures (right).

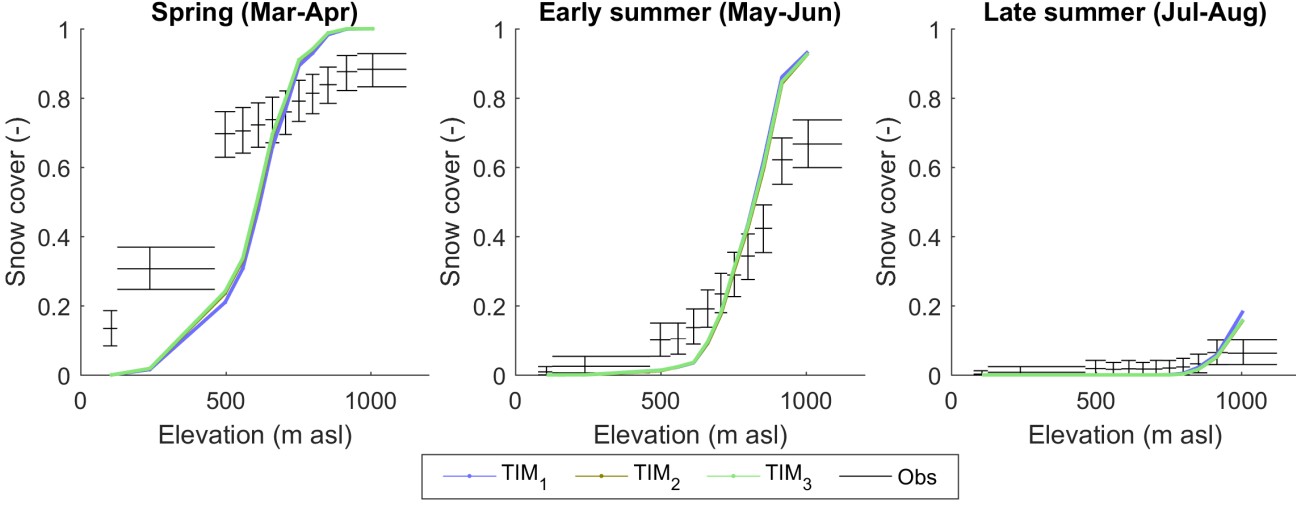

**Figure 12.** Simulated seasonal snow distribution curves when using the three melt model structures.





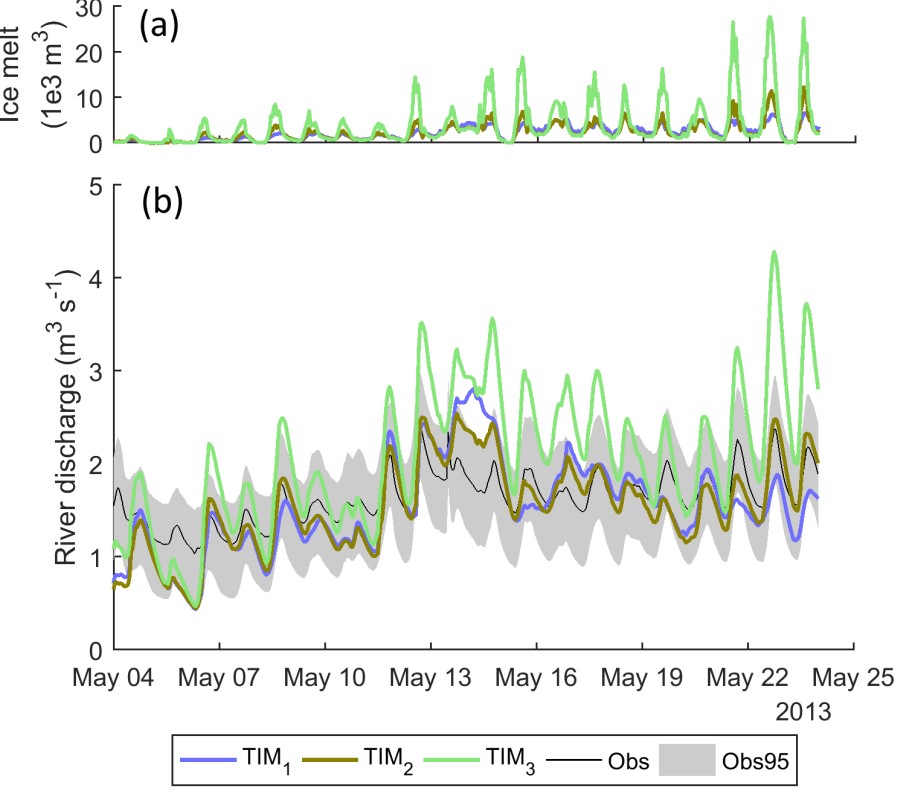

**Figure 13.** Mean simulated hourly ice melt (a) and river discharge (b) during May 2013 using the top 1% of models from the three melt model structures in combination with the ROR$_1$ runoff routing model structure.

signatures. The simplest GHM configurations using the TIM$_1$ and TIM$_2$ model structures capture 12 river discharge signatures simultaneously within the LOA while the inclusion of the dynamic snow albedo term and re-arrangement of the melt equation in the TIM$_3$ melt model actually inhibits the GHM performance where only 10 of the 21 river discharge signatures are captured within the LOA.

5    The mean monthly flow signatures for January, February and May show some of the highest absolute acceptability scores indicating the models are least efficient at capturing these. For winter flows in January and February, the simulation using the TIM$_2$ model structure is substantially more acceptable than when using the other melt model structures although it should be noted that, given that flows are very low here, the absolute error is less than 0.2 m$^3$ s$^{-1}$. For the mean May river flow, the simulation using TIM$_3$ is less acceptable than when using the other melt model structures. This is interesting, as May

10  coincides with the beginning of the main melt season and therefore this could be some indication of an inability to capture this initialisation properly. A comparison of the simulated ice melt during May 2013 reveals that the TIM$_3$ structure simulates the highest ice melt of all three melt model structures (Fig. 13a) which results in a positively-biased river flow time-series (see Fig. 13b). It has already been shown that the simulated snow coverage signatures are almost identical when using the





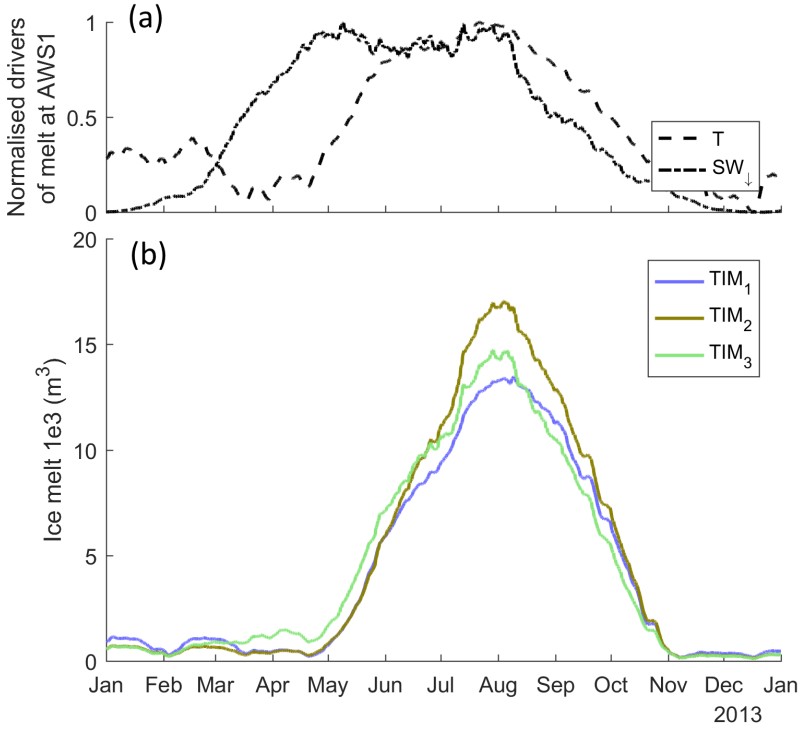

**Figure 14.** Normalised temperature and incident solar radiation (a) and simulated ice melt from the three calibrated ice melt model structures (b) for the year 2013. All time-series use a monthly moving average filter.

three melt model structures and as such the contrast in simulated ice melt cannot be directly attributed to the dynamic snow albedo component of $TIM_3$. A comparison of the simulated ice melt time-series over 2013 with a monthly moving-average filter reveals that the positive melt bias from $TIM_3$ extends between April and June (Fig. 14b) which corresponds to the period where temperatures are relatively low, but where incoming solar radiation is relatively high (see Fig. 14a). As such, it is the additive form of the $TIM_3$ melt equation and the subsequent increased influence of solar radiation on melt which induces the bias in flow simulations in the early melt season.

Of the remaining river discharge signatures, only a handful show any substantial difference when switching between the melt model structures including the mean April and August discharge and the two 'flashiness' signatures: the integral time and the rising limb density. However, the differences here are very small. For the 'high slope' signature, which characterises the variability of high flow river flows, the simulation using the $TIM_1$ melt model structure is able to capture it within the LOA, while the simulations using the $TIM_2$ and $TIM_3$ model structures both show a negative bias suggesting they underestimate high flow variability. The reason for this is not clear, but it does indicate that the simpler $TIM_1$ melt model structure is better suited for capturing this signature.



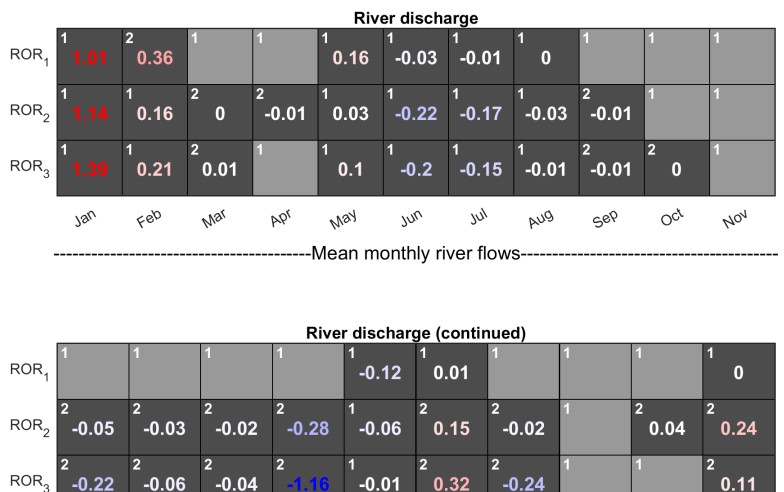

**Figure 15.** Acceptability scores obtained after calibrating the GHM using the three runoff-routing model structures in combination with the TIM$_1$ melt model structure. Light grey boxes indicate acceptable simulations ($s = 0$) and numbered, dark-grey boxes indicate unacceptable simulations coloured blue and red to indicate negative and positive bias respectively. White numbers in top left of each box indicate relative ranking where acceptability scores are substantially different between the GHM configurations.

### 3.3 Acceptability of runoff-routing model structures

To evaluate the runoff-routing model structures, acceptability scores have been calculated for the river discharge signatures only as these structures do not influence ice melt or snow coverage (Fig. 15). To ensure fair comparison between the different structures, all scores have been obtained using the simplest TIM$_1$ melt model structure in the GHM.

5    It was noted previously, that all melt model structures used in combination with ROR$_1$ resulted in positively-biased January and February river flows. It could be that including a more complex non-linear runoff-routing model structure in the GHM could help to mitigate this bias. Indeed, the calibrated simulations do show a substantial reduction in positive bias for the mean February flows when using ROR$_2$ and ROR$_3$, however the simulations are still unacceptable. Furthermore, for the mean January river flow there is no substantial change in acceptability score. This indicates that the runoff-routing representation is also not the reason for this overestimation of flows at the beginning of the year. To investigate this positive bias further, Fig. 16c shows the simulated time-series from the calibrated models using TIM$_1$ in combination with ROR$_1$, ROR$_2$ and ROR$_3$ for January and February 2013. Figure 16a shows that melt is an insignificant input during these winter months (green line). Rather it is rainfall (black dash) that dominates the runoff input and this results in two pronounced peaks in the simulated river discharge time-series. The different behaviour of the simulations using the three runoff-routing model structures is much more





obvious during the rainfall-runoff events. The simulation using the $ROR_1$ structure is noticeably more flashy in response to the rainfall and overestimates the peak flows while the $ROR_2$ and $ROR_3$ simulations, which include additional, more diffusive representations of the flow of water through snow and firn, result in peak flows that are closer to the observed, but with a recession that is too shallow. Regardless of these deficiencies, however, all result in an almost identical positive bias as shown

by the cumulative flow in Fig. 16b. The more complex routing structures, therefore, appear to offer no additional capabilities to correct the positive monthly flow biases.

There are however differences when assessing other aspects of the river discharge time-series, particularly in the signatures relating to high flows. In Fig. 15, it can be seen that while the simulation using the $ROR_1$ routing model structure is able to capture all of the high flow signatures simultaneously, the $ROR_2$ and $ROR_3$ structures show an unacceptable negative bias for

these signatures indicating underestimation of high flow magnitude and variability. To evaluate this in more detail, Fig. 16f shows the simulated time-series for the highest recorded river flow event during October 2014. Here, the flashier and more responsive $ROR_1$ structure achieves the closest fit to the observed peak flow and within the uncertainty bounds while the more diffusive, $ROR_2$ and $ROR_3$ structures underestimate the peak flow. Note they also underestimate the overall river flow variability as indicated by the coefficient of variation signature. It appears, therefore, that the diffusive behaviour of the $ROR_2$

and $ROR_3$ runoff-routing structures which is advantageous for capturing peak flows in winter, results in underestimation of peak flows at the end of the melt season and an underestimation of overall river flow variability.

### 3.4 Consistency of melt model structures

The results so far have highlighted some inconsistencies in the GHM configurations using the melt and runoff-routing model structures where they are unable to reconcile some combinations of signatures simultaneously. This is important as those

inconsistencies could help to further diagnose structural deficiencies in the different model structures. To investigate this, consistency scores have been calculated between pairs of the 33 signatures for each GHM configuration. A model can be deemed consistent across a pair of signatures if it is able to capture both within their LOA simultaneously. The consistency scores are therefore calculated as the minimum sum of the two acceptability scores between a pair of signatures across the 5000 calibration runs for each GHM configuration.

Figure 17 shows the average consistency scores calculated across the signatures for each attribute of ice melt, snow coverage and river discharge using the three melt model structures in combination with the $ROR_1$ runoff-routing structure. The top panel shows the consistency scores when using the simplest $TIM_1$ melt model structure. The regions in red highlight the areas where the GHM is inconsistent. The first striking observation is the red band along the upper catchment snow coverage attribute. It has already been demonstrated that the simulations using the $TIM_1$ structure cannot reconcile the upper-catchment snow coverage

with the remaining snow coverage signatures. This further demonstrates that when using the $TIM_1$ structure, the GHM cannot reconcile the upper-catchment snow coverage with any of the other attributes.

The largest inconsistency score obtained was between the short term, seasonal melt on the glacier tongue and long-term total glacier volume change. This is further evidence that the $TIM_1$ melt model structure is not able to reconcile the melt signatures over the differing temporal and spatial scales. This raises the question of where this inconsistency stems from. It



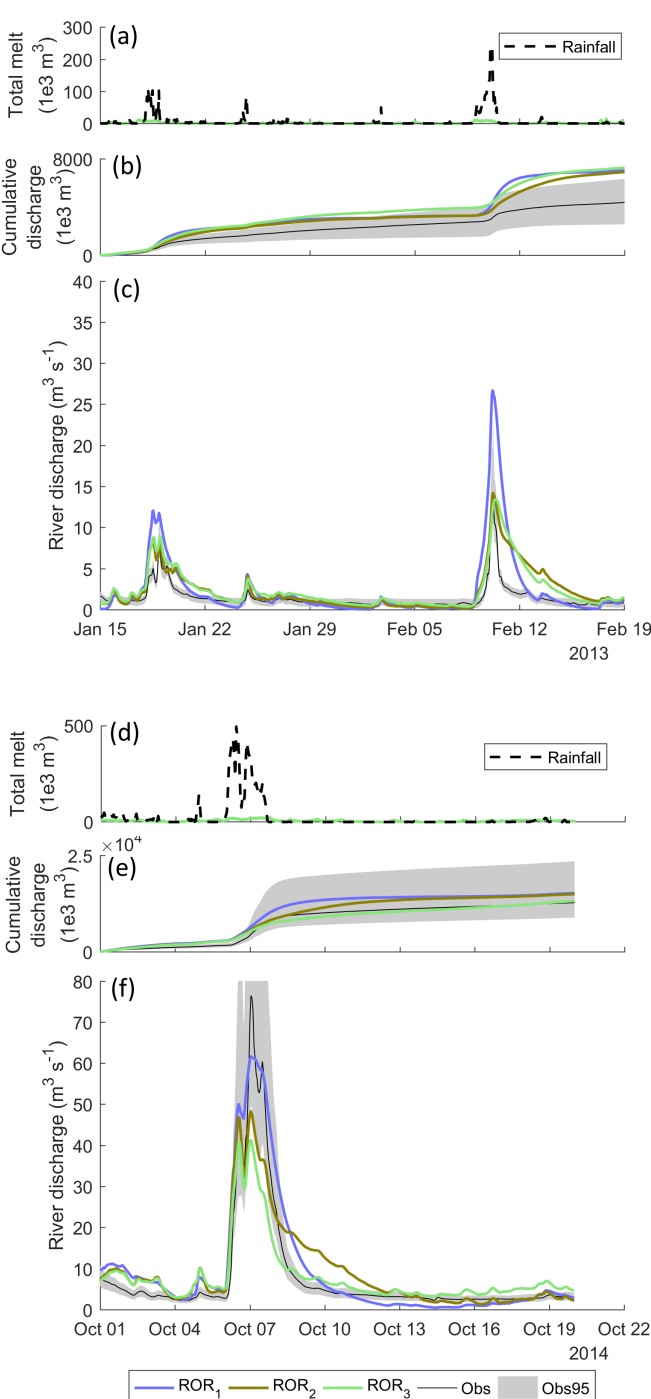

**Figure 16.** Simulation time-series using the three different runoff-routing model structures in combination with the TIM$_1$ melt model structure including simulated total melt and rainfall (top), cumulative river discharge (middle) and river discharge time-series (bottom) for January and February 2013 (a,b,c) and the October 2014 flood (d,e,f).





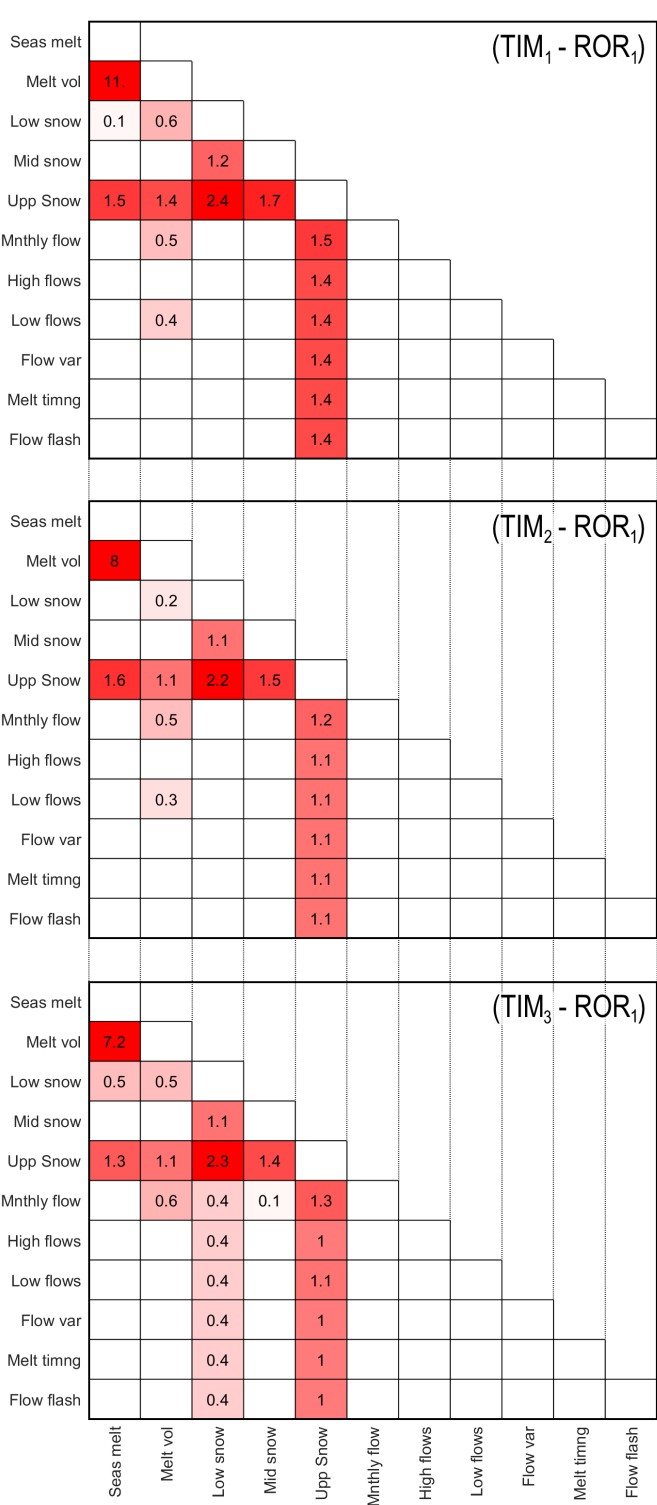

**Figure 17.** Average consistency scores between attributes using the three melt model structures in combination with the $ROR_1$ runoff-routing structure. Scores of < 0.1 have not been reported.



should be noted that the seasonal melt signatures show a small inconsistency with the lower-catchment snow coverage and a larger inconsistency with the upper-catchment snow coverage. The total glacier volume change signature, however, is also inconsistent with the monthly flow and low flow signatures indicating that it is the long-term glacier wide mass balance that the model is getting wrong.

The use of the $TIM_2$ model structure which includes topographic effects goes some way to reducing most of the inconsistencies shown using the $TIM_1$ model structure (Fig. 17 middle panel). However, all but one of the inconsistencies (between lower-catchment snow coverage and seasonal melt) remain, indicating that the use of the $TIM_2$ melt model structure only provides a small improvement in model consistency.

Using the $TIM_3$ model structure also helps to improve model consistency, particularly those associated with the upper snow coverage, but surprisingly it also introduces new inconsistencies in relation to the lower-catchment snow coverage, where the model is not able to reconcile these signatures with any of the other attributes. This is interesting, because although the simulated snow distribution curves from each melt model structure were approximately identical, the alteration of the melt equation and inclusion of dynamic snow albedo reduces overall model consistency.

### 3.5 Consistency of runoff-routing model structures

Consistency scores have also been calculated for each pair of river discharge signatures (Fig. 18) using the three runoff-routing structures in combination with the $TIM_1$ melt model structure. The simulations using the $ROR_1$ structure (top panel) and next simplest $ROR_2$ structure (middle panel) show a very similar pattern of model inconsistencies. Firstly, both sets of simulations do not capture the relatively low flows in February and the relatively high flows in July and August simultaneously. This corroborates the findings from the acceptability analysis which revealed a tendency for the model structures to overestimate low flows in the winter and underestimate high flows in the summer and autumn, particularly with relation to rainfall-induced high flows. Interestingly though, the seasonal flow inconsistency is centred on February and there are not inconsistencies for the other low flow months from January to April. This provides further evidence that it is particularly the rainfall-induced flows that the model is not able to capture effectively. In fact, February has some of the highest flows in the record of winter flows induced by large rainfall events (see average flow signatures in Table 1). This suggests this could be the reason that the inconsistencies between winter and summer flows are centred around these months. The inclusion of additional flow pathways in the routing routine only enhances these inconsistencies, particularly when using the $ROR_3$ model structure where the inconsistencies extend into June (bottom panel).

The $ROR_1$ simulations show inconsistencies between the February flows and low flow variability as indicated by the low slope signature. The reason for this is not clear, but interestingly, the inclusion of an extra, more diffuse, flow pathway in the $ROR_2$ model appears to remedy this, suggesting that there is some non-linear behaviour that the $ROR_1$ model structure cannot capture. However, it comes at the cost of inducing an extra inconsistency between the mean flows in January and the overall flow variability as indicated by the coefficient of variation. This new inconsistency is amplified when using the $ROR_3$ structure. This further corroborates the findings from the acceptability scores where the more complex runoff-routing model structures cannot capture the observed flow variability adequately.





Interestingly, the consistency scores when using the $ROR_1$ and $ROR_2$ structures are relatively similar, with each configuration demonstrating inconsistencies between four and five pairs of river discharge signatures respectively. In contrast, using the most complex $ROR_3$ structure introduces a number of new inconsistencies with a total of 12 inconsistent pairs of simulated river discharge signatures. These new inconsistencies are centred around the mean monthly flow signatures as well as the signatures

relating to high and low flow magnitude and variability.

## 4    Discussion

The first aim of this study was to investigate if a signature-based approach within a LOA framework could be used to diagnose deficiencies in the different melt and runoff-routing model structures. The comprehensive set of signatures provided a powerful method to evaluate the model behaviour. Furthermore, when used within the LOA framework, it was straightforward to identify

those aspects of the glacio-hydrological system that the GHM configurations could not capture. A number of the identified model deficiencies are particularly important in the context of future river flow predictions which will now be discussed.

Regardless of the choice of melt model structure, all GHM configurations were able to capture the three signatures of ice melt individually, but none of them could capture all of the signatures simultaneously. The challenge here was to reconcile three signatures that characterise glacier melt over different spatial and temporal scales. This is not a straightforward task,

particularly when using temperature index models that lump a number of spatially and temporally variable terms from the full energy balance equation into a handful of calibration parameters which may lack robustness in space and time (MacDougall et al., 2011; Matthews et al., 2015; Gabbi et al., 2014). The inclusion of solar and topographic effects in the $TIM_2$ and $TIM_3$ melt model structures addressed some of these limitations. Indeed, the inclusion of these in conjunction with the dynamic snow albedo parameterisation returned the most acceptable simulations of the ice melt signatures overall. However, further

improvements are required to achieve acceptable model simulations that capture all of the ice melt signatures simultaneously. Certainly, one aspect of the glacier which was not accounted for was debris cover at the glacier terminus which could be an important control on point scale and overall ice mass balance. Some TIMs that include representations of debris cover do exist (e.g. Carenzo et al., 2016) and the signature-based LOA approach would provide the ideal framework for evaluating the added-value of further structural modifications like these.

The snow coverage signatures highlighted deficiencies in all of the GHM configurations. None of the prior 45,000 model compositions were able to capture the spring and early-summer snow coverage in the upper catchment and all of the calibrated GHM configurations overestimated snow coverage in the upper catchment whilst underestimating it in the lower catchment. Interestingly, using the most sophisticated $TIM_3$ structure with the dynamic snow albedo function had almost no effect on the overall acceptability across these signatures, indicating that the melt model formulation was not the primary source of

model deficiencies here. Of course, snow coverage simulations are sensitive to other components of the GHM such as the snow redistribution model, which itself, is sensitive to the resolution of the DEM used to parameterise it; a coarser DEM resolution removes peaks and troughs in the land surface which can bring about more complex patterns of snow coverage. Similarly, the glacier volume change signature will be sensitive to the glacier evolution formulation and parameterisation. It is clear,





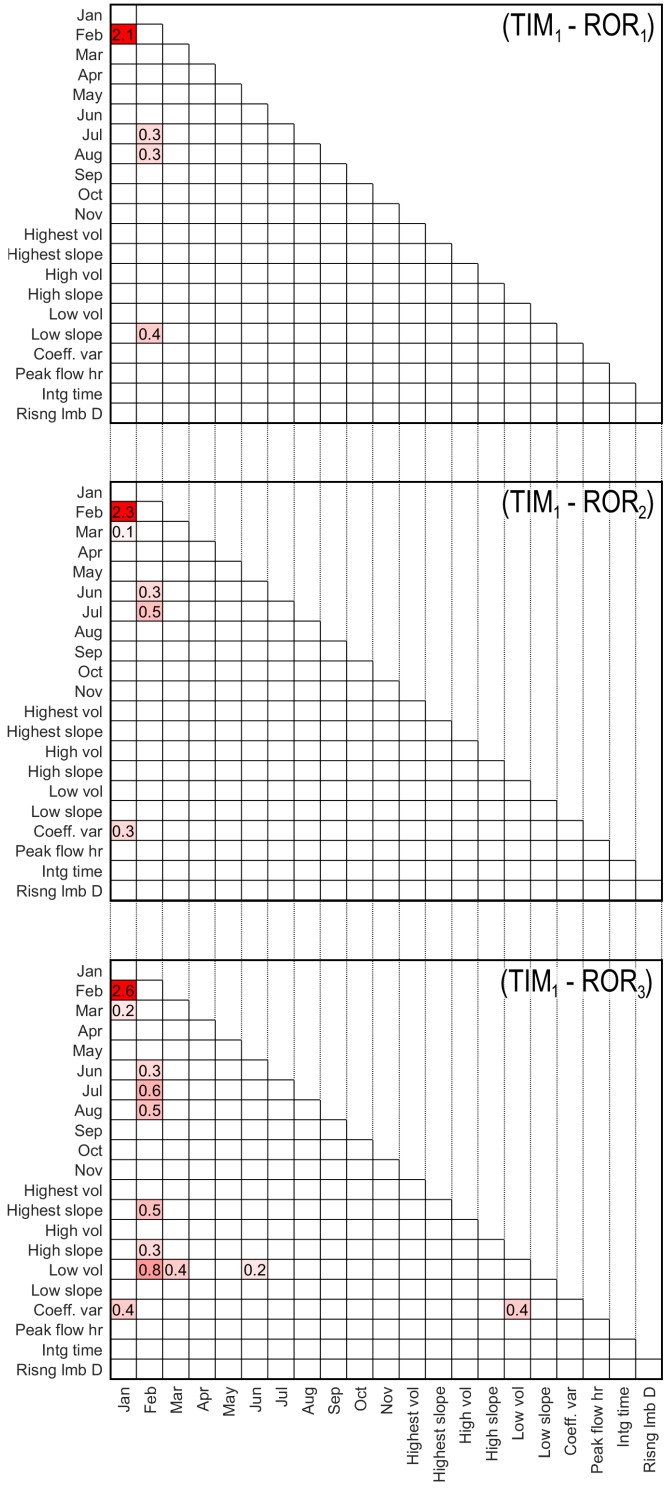

**Figure 18.** Average consistency scores between river discharge signatures using the three runoff-routing model structures in combination with the $TIM_1$ melt model structure.





therefore, that while the application of the LOA framework here has demonstrated the gains that can be made in capturing some signatures through the inclusion of extra model complexity, the apparent insensitivity of the snow coverage signatures to structural modifications indicates that further gains may also be made by investigating other components of the GHM structure within this framework. Indeed, beyond the structural nature of a GHM, the boundary conditions used may also contribute to

model deficiencies. In particular, the driving precipitation data, which while relatively well constrained by observations at the bottom of the catchment, are less certain higher up in the catchment where limited observation data are available. Indeed, one could explain the tendency to overestimate snow coverage higher up in the catchment by a positive bias in the driving precipitation data here. Such a bias could also explain the modelled inconsistencies across the ice melt signatures. However, before making such a conclusion, it should be noted that the limited precipitation collected at the summit of Öræfajökull

indicate that the driving precipitation data used in this study are approximately correct (Guðmundsson, 2000). Even so, the recent melt model comparison by Reveillet et al. (2017) suggest that uncertainties in driving precipitation data can cloud any differences between melt model behaviour.

As a side experiment, we tried increasing the snow melt parameters for the $TIM_3$ model to see if this would help remove some of the inconsistencies across the ice and snow signatures. It was found that using snow melt parameters equal to or even

greater than those used for the ice reduced (improved) the consistency score between the ice melt signatures by more than 50% when using $TIM_3$. The physical justification of this is of course questionable, but it does highlight the influence that the prior parameter distributions could have on the results presented here. Accordingly, different specifications of the calibration parameter ranges may also help to improve model consistency.

The main deficiencies noted for all of the GHM configurations when compared to the river discharge signatures were an

overestimation of the relatively low winter flows in January and February, and the flows at the start of the melt season in May. It was assumed that the addition of extra 'slow' flow pathways in the $ROR_2$ and $ROR_3$ runoff-routing structures would help to correct for any deficiencies in capturing the hydrograph seasonality. Instead, the choice of runoff-routing structure had very little influence on these signatures, indicating that longer term storages of water do not have a major control on the seasonality of the hydrograph. This is probably because of the catchment's small size which leads to an instantaneous seasonal response

to melt on a monthly timescale. We suggest, however, that for larger catchments, the monthly flow signatures are likely to be more sensitive to the choice of runoff-routing structure. Instead, the simulated mean monthly river flow signatures were more sensitive to the choice of melt model structure, particularly in May at the start of the melt season, which is not surprising given the high degree of glaciation of the river basin. Even so, regardless of the melt model structure employed, none of the GHM configurations could correct for the biases in mean monthly flows indicating that further structural modifications are

required. One process that is not represented at all in any of the GHM configurations, but which has shown to be important in for Icelandic glaciers is refreezing of melt water and rainfall (Johannesson et al., 1995). It is estimated that about 7% of total melt in valley glaciers in Iceland refreezes, and therefore, the inclusion of this process could also help to reduce runoff during the colder months of January and February.

In contrast to the monthly river flow signatures, the choice of runoff-routing structure had by far the dominant control on

those signatures that are controlled by flows operating on much shorter timescales such as the distribution of flows, flow





variability and flashiness. This hierachy of influence between the melt and runoff-routing model structures has important implications for river discharge prediction uncertainty in glaciated basins. For example, if one were interested in future seasonal water resource availability, they would be most reliant on predictions of mean monthly river flows. The results here indicate that, for this catchment at least, uncertainties in these predictions stem primarily from melt model uncertainty. In contrast if one

were interested in future changes in flood frequency, the dominant source of model prediction uncertainty is the runoff-routing approach. Uncertainties in river flow predictions from glacio-hydrological models are therefore dependent on the river flow characteristic of interest.

The results from the simulated river discharge signatures also raised some questions about the added value of introducing extra complexity to conceptual models of glacio-hydrological processes. The most sophisticated $TIM_3$ melt structure was the

most consistent across the ice melt and snow coverage signatures. However, it was also the least acceptable structure for the mean May river flow signature, an artefact of its additive form. Similarly, the $ROR_3$ structure, originally proposed as the most realistic conceptual representation of water storage and transmission in the river basin, was the least acceptable model across the river discharge signatures. These results highlight the need to exercise caution before introducing complexity to conceptual models of glacio-hydrological processes. They also illustrate the importance of testing prior assumptions about the system

against other possible model hypotheses, for which the signature-based LOA framework is ideally suited.

The second aim of this study was to determine if the signature-based evaluation within a LOA framework could be used to constrain the prior population of model structures and parameter sets (compositions) down to a smaller population of acceptable models. The initial discrimination tests showed that all of the signatures have discrimination power, although for two of the snow signatures, none of the 45000 model compositions could capture them. The mean January and May river flow sig-

natures were the best discriminators, individually reducing mean river discharge uncertainty to 60 - 70% of that from the full population of model compositions, although it should be noted that the majority of this reduction stemmed from constraining the acceptable parameter sets rather than the model structures. These results indicate that the LOA framework could be used to find a population of acceptable model compositions. However, the fact that none of the prior 45000 compositions were able to capture all of the signatures means that this remains to be seen. At a fundamental level, the results indicate that the structural

configurations of the GHM employed in this study are simply not good enough to capture the observation data within their observation uncertainty bounds. To address this, one could implement further structural modifications, some of which have been alluded to, until acceptable simulations are obtained. Indeed, a more thorough exploration of a wider parameter space could also yield acceptable model compositions. There are of course other sources of unacceptable behaviour though. Of these, boundary conditions including the ice and watershed boundaries as well as the driving climate data are all contenders. The

initial ice geometry was also uncertain but not explicitly accounted for. It is therefore recommended that where possible, future applications of the LOA framework should incorporate these additional sources of uncertainty, so that more robust conclusions about model appropriateness can be made. Certainly, study sites with good observation data and understanding of data uncertainty would be ideal candidates for these future applications.





## 5  Conclusions

The signature-based, LOA framework adopted in this study provided a comprehensive evaluation of different GHM melt and runoff-routing model structures. In contrast to traditional model evaluation approaches which rely on one or several global summary statistics, the adoption of multiple signatures helped to identify those aspects of the glacio-hydrological system that a particular model could or could not capture and the added value of introducing additional complexities to simplified process models. When evaluated against individual signatures, the more complex model formulations did improve model simulations in some cases. However, they were not necessarily more consistent across the full range of signatures, emphasising the need to exercise caution and properly evaluate if additional complexities are justified. The often conflicting acceptability scores across the signatures highlights the difficulty and inherent uncertainty in model structure selection. It is clear, therefore, that future glaciological and hydrological projection studies that use simplified model structures should take account of these uncertainties, although to date these have rarely been considered. For future river flow predictions in glaciated basins it is likely that the source of model uncertainty depends on the particular river flow characteristic of interest. We found evidence that a hierarchy of influence exists between the melt and runoff-routing model structures across the range of river discharge signatures.

An additional advantage of adopting the LOA framework is that it provides objective criterion for accepting or rejecting particular model structures and parameterisations. While all, but two, of the signatures demonstrated discrimination power, none of the 45,000 different model compositions tested in this study were able to capture them within their LOA simultaneously. Therefore, it remains to be seen if the framework can be used in this way, although we suggest that applications that go beyond examining the melt and runoff-routing structural uncertainties may prove more fruitful in obtaining a behavioural population of models. These should consider other uncertainties including those associated with snow redistribution, glacier evolution and model boundary conditions. We would therefore encourage future studies, particularly where a broad range of observation data covering different aspects of the glacio-hydrological system, to move away from using traditional global summary statistics for model evaluation and adopt a multi-metric approach within a LOA framework so that their simplified process hypotheses can be rigorously tested, and structural uncertainty better understood.

## Appendix A:  Glacio-hydrological model

### A1  Soil infiltration and evapotranspiration

The semi-vegetated nature of the catchment coupled with the relatively cool temperatures year-round mean that evapotranspiration is generally low (Einarsson, 1972). Even so, to satisfy the water balance, an explicit representation of the soil zone for model nodes that are not ice or snow-covered was included using the method developed by Griffiths et al. (2006) which has been successfully applied to temperate regions in the past (Mackay et al., 2014; Sorensen et al., 2014) and is based on the well established UN Food and Agricultural Organisation soil water balance method (Allen et al., 1998). For each bare ground node, the soil is represented as a finite storage reservoir with a soil water capacity, termed the total available water, $TAW$ [L], which





defines the maximum volume of water available to plants for evapotranspiration after the soil has drained to its field capacity and can be defined from lookup tables with basic information on vegetation and soil information (Allen et al., 1998). This was parametrised using the 'Talus' soil class and 'semi-vegetated' land surface class giving an average $TAW$ value of 7 mm. Soil storage is replenished by infiltration from rainfall and melting of residual snow overlying the bare ground and is depleted by evapotranspiration giving a soil water balance:

$$\frac{\Delta S_{soil}}{\Delta t} = I - ET \tag{A1}$$

where $S_{soil}$ [L] is the soil water storage, $t$ is time, $I$ [LT$^{-1}$] is the infiltration rate and $ET$ [LT$^{-1}$] is the evapotranspiration rate. Because measured $ET$ is rarely available, Griffiths et al. (2006) propose using the potential evapotranspiration rate, $ET_0$, instead which defines the evapotranspiration rate from a reference grass covered wet soil (see Appendix A2 for calculation of $ET_0$). Using $ET_0$ as the maximum possible evapotranspiration rate, they define a separate function which accounts for the fact that as the soil becomes drier, plants find it more difficult to extract moisture from the soil matrix, and therefore $ET$ is typically less than $ET_0$. While this is conceptually sound, it was decided not to include this function and instead assume that $ET = ET_0$. There are three reasons for doing this. Firstly, because the inclusion of this function requires an additional parameter which is uncertain and must be calibrated. Secondly because $ET$ is a relatively small component of the overall water balance in this catchment and it was not the aim of this study to investigate this aspect of the catchment hydrology. Thirdly, because previous studies have shown that this parameter (and therefore the behaviour of this function) is relatively insensitive and unidentifiable (Mackay et al., 2014).

In the original formulation by Griffiths et al. (2006), any excess soil water (i.e. when $S_{soil} > TAW$) is distributed between overland flow and groundwater recharge pathways. They use a fixed baseflow index ($BFI$) parameter which defines the proportion of soil water excess that recharges the groundwater. Given the nature of the Virkisá river basin (thin soils overlying impermeable bedrock), it was assumed that soil water migrates to the river outlet via relatively fast, overland flow pathways only and so the $BFI$ parameter was set to zero.

## A2    Potential evapotranspiration

Potential evapotranspiration can be calculated from measured meteorological data, most simply as a linear function of measured temperature (e.g. Blaney and Morin, 1942), or where measurements of windspeed, air pressure and solar radiation exist, the full Penmen-Monteith combination equation can be solved. Given that these additional variables are measured at AWS1 from 2009, the combination equation as defined by Allen et al. (1998) was used to calculate hourly potential evapotranspiration over this period:

$$ET_0 = \frac{0.408\Delta(R_n - G) + \gamma\frac{900}{T+273}u(e_s - e_a)}{\Delta + \gamma(1 + 0.34u)}h \tag{A2}$$

where $ET_0$ is the daily average potential evapotranspiration rate (mm d$^{-1}$), $R_n$ is the net radiation (MJ m$^{-2}$ d$^{-1}$), $G$ is the soil heat flux (MJ m$^{-2}$ d$^{-1}$), $e_s$ and $e_a$ are the saturation and actual vapour pressure respectively (kPa), $\Delta$ is the rate of change of the





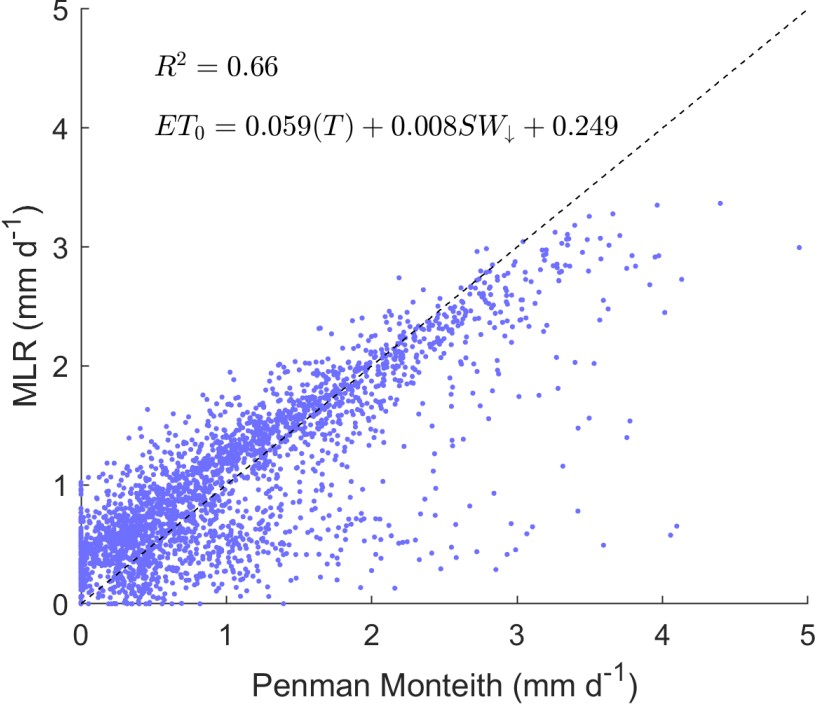

**Figure A1.** Multiple linear regression model used to convert ambient air temperature and incoming solar radiation into potential evapotranspiration.

saturation vapour pressure with temperature (kPa $^{\circ}$C$^{-1}$), $\gamma$ is the psychrometric constant (kPa $^{\circ}$C$^{-1}$), $u$ is the wind speed (m s$^{-1}$) and $T$ is the mean daily ambient air temperature ($^{\circ}$C).

Prior to 2009, the viability of using $T$ as a proxy for $ET_0$ in a linear regression model framework like Blaney and Morin (1942) was investigated. Similarly, incident solar radiation was also used as the independent variable for this model. In fact, the best fit was achieved using both variables in a multiple linear regression model which was able to explain 66% of the $ET_0$ variance (Fig. A1). This model was used to distribute $ET_0$ in space and time using the driving temperature and incident solar radiation data.

## A3   Glacier geometry evolution

The empirical $\Delta$-h parametrisation (Huss et al., 2010) requires the availability of at least two digital elevation models of the glacier separated in time. The difference between the two is used to define the $\Delta$-h polynomial which has the form:

$$\Delta h = (h_r + a)^{\gamma} + b(h_r + a) + c \tag{A3}$$



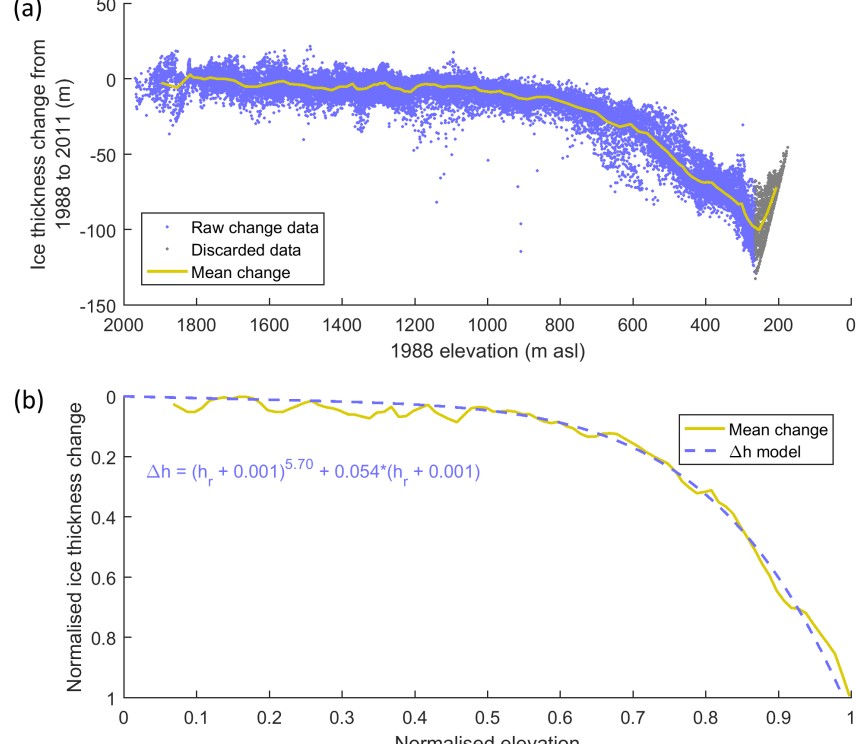

**Figure A2.** Raw elevation change data from 1988 and 2011 ice DEMs (a) and fitted $\Delta h$ model to normalised mean elevation change curve following Huss et al. (2010) (b).

where $\Delta h$ is the normalised surface elevation, $h_r$ is the normalised elevation range and $a$, $b$, $\gamma$ and $c$ are fitted parameters. For this study, the two digital elevation models from 1988 and 2011 were used to define this relationship. Figure A2 (top) shows the raw change data against the 1988 ice elevation. It was decided that the data at the very front of the glacier should not be used as here the ice has completely melted and as such the bedrock beneath skews the raw change data. Figure A2 (bottom) shows the fitted $\Delta h$ model to the normalised mean elevation change curve. Following Huss et al. (2010), the glacier geometry is updated each year by distributing the net glacier mass balance across the glacier according to this relationship.

**Appendix B:  Temperature lapse rates**

In order to investigate seasonal variations in lapse rate, the temperature gradient between the lowest (AWS1) and highest (AWS4) weather stations in the Virkisá river basin were analysed. The results showed a remarkable degree of variation in hourly average lapse rate between the months of the year (white lines in Fig. B1). During the winter months between November and February, the lapse rate is a relatively stable -5 ºC km$^{-1}$ throughout the day. In contrast, between March and October there is a pronounced diurnal variation in lapse rate where it is strongest in the late afternoon/early evening. The heat maps in Fig. B1



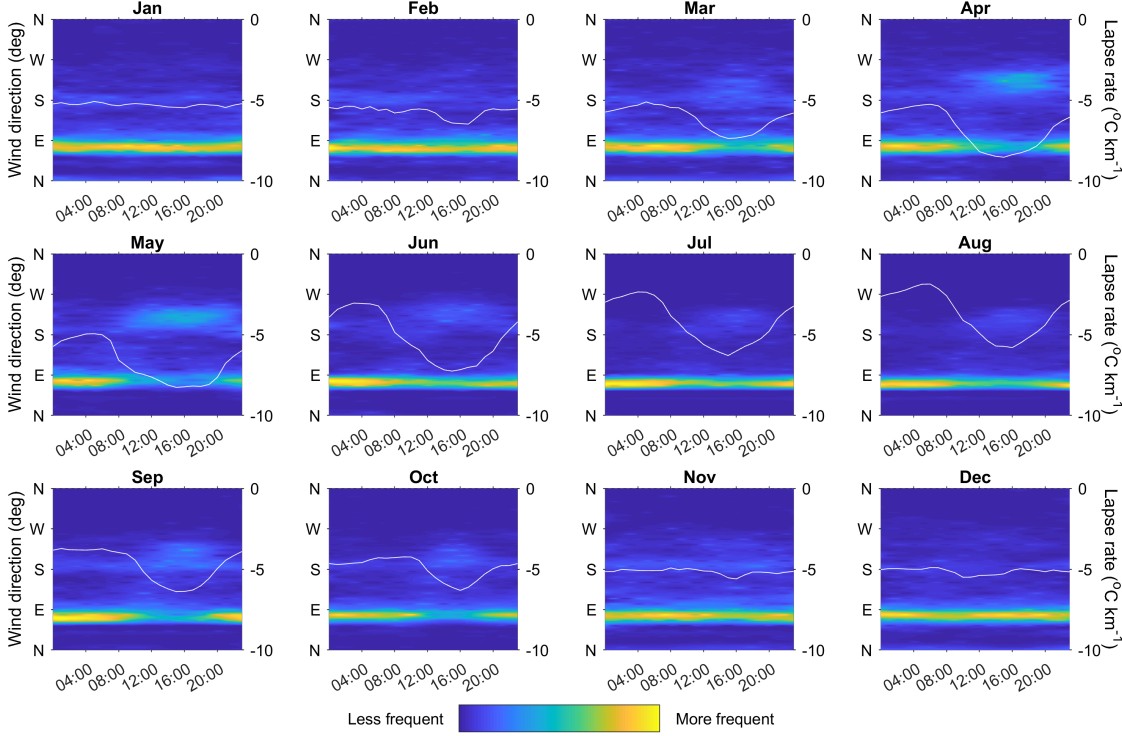

**Figure B1.** Monthly average hourly temperature lapse rates (white lines, right hand axis) derived from AWS1 and AWS4 temperature time-series overlying heat maps which represent the frequency distribution of hourly wind direction data from AWS4 (left hand axis).

represent the frequency distribution of wind direction for each month and show that the development of the strongest lapse rates in the afternoon correspond with a break up of the prevailing northeast winds that flow down from the summit of Öræfajökull and a switch to winds from the southwest. Petersen and Pellicciotti (2011) found a similar phenomenon on the Juncal Norte Glacier in the semi-arid Chilean Andes. They attributed the shallow temperature gradient in the morning with katabatic winds

5    flowing down glacier which serve to cool the air over the glacier and weaken the lapse rate. In the afternoon, they showed that a breaking up of this layer by valley winds served to increase the temperature gradient by warming the air over the lower glacier. This suggests that winds flowing down from the Öræfajökull summit in the warmer months could serve to cool near surface air temperatures over the ice, thereby retarding ice melt. To account for this phenomenon, Petersen and Pellicciotti (2011) suggest adopting the Shea and Moore (2010) model to correct on-ice temperatures relative to ambient off-ice weather station

10   measurements. Shea and Moore (2010) found that for three glaciers on the southern Coast Mountains of British Columbia, Canada, there was a threshold in ambient off-ice air temperature, above which the winds flowing over the glacier served to cool the near-surface on-ice air. They suggest this temperature lies somewhere between 4 and 8 °C, but is likely to be site specific.





To investigate if such a threshold exists on the Virkisjökull glacier, five Gemini Tinytag Aquatic 2 temperature loggers were deployed across the glacier at elevations ranging from 150 - 400 m asl. Each logger was secured at 1.5 m above the ice in a white PVC radiation shield attached to a tripod (Fig. B2). The sensors were deployed for 7 days in late August 2016 and then for a further 7 days in early March 2017 to represent summer and winter on-ice temperatures respectively. The loggers were

synchronised in time with the AWS weather stations and set to measure temperature every 15 minutes. This allowed for the direct comparison of on and off-ice near surface temperatures.

Figure B3 shows the synchronised on and off-ice temperatures from all of the measurements taken in winter (blue dots) and summer (yellow dots). The off-ice temperatures were derived assuming a linear lapse-rate between AWS1 and AWS3 as these are situated at elevations similar to the Tinytag temperature loggers. The results show that there is a temperature threshold

above which on-ice temperature falls below off-ice temperature which was estimated to be 5.27 $^\circ$C. Following Petersen and Pellicciotti (2011); Shea and Moore (2010); Ragettli et al. (2014), this cooling effect was interpreted as being due to northeast winds which bring cooler air from above over the tongue of the glacier, thereby cooling the on-ice air temperature and the piecewise function derived from Fig. B3 was employed to correct temperatures on the ice during the warmer months when ambient air temperatures exceed this threshold:

$$T_{on} = \begin{cases} T_{off} & T_{off} \leq 5.27 \\ 0.74 \cdot T_{off} + 1.38 & T_{off} > 5.27 \end{cases} \qquad \text{(B1)}$$

where $T_{on}$ and $T_{off}$ are the on and off-ice near-surface air temperature ($^\circ$C).

## Appendix C: Calibration parameters

Table C1 lists all of the calibration parameters for the melt and runoff-routing model structures which were randomly perturbed during the GHM calibration procedure.

*Author contributions.* JDM ran all model experiments and conducted the analysis of results. He also led the writing of this manuscript. All co-authors contributed to formulation and discussion of methods used as well as writing of manuscript.

*Competing interests.* The authors declare they have no competing interests.

*Acknowledgements.* This work was supported by a NERC studentship awarded to JDM via the Central England NERC Training Alliance (CENTA). The authors' acknowledge the support of Joaquin Maria Munoz Cobo Belart (University of Iceland) for providing the 1988 ice

DEM of Öræfajökull as well as Dr Andrew Black (University of Dundee) and Lee Jones (British Geological Survey) for providing the river discharge and terrestrial LIDAR data used in this study. We would also like to acknowledge the useful discussions with Prof Jim Freer and Dr





**Figure B2.** Example of Gemini TinyTag housing used for measuring on-ice temperature at one location on ice.





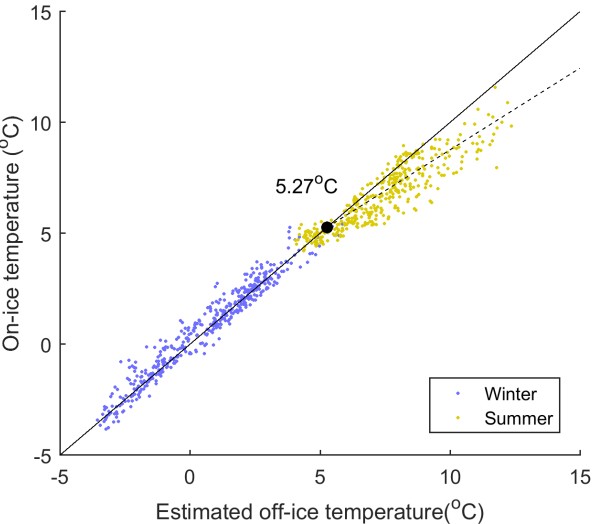

**Figure B3.** Derived temperature threshold where on-ice temperature is cooler than the ambient off-ice temperature using Shea and Moore (2010) model.

**Table C1.** Calibration parameters for the melt and runoff-routing model structures.

| Structure | Parameter | Calibration range | Units |
|---|---|---|---|
| TIM$_1$ | a$_{ice}$ | 2.0e-4 - 7.0e-4 | m we $^o$C$^{-1}$ hr$^{-1}$ |
| | a$_{snow/firn}$ | 4.0e-7 - 2.0e-4 | m we $^o$C$^{-1}$ hr$^{-1}$ |
| TIM$_2$ | a$_{ice}$ | 2.0e-4 - 7.0e-4 | m we $^o$C$^{-1}$ hr$^{-1}$ |
| | a$_{snow/firn}$ | 4.0e-7 - 2.0e-4 | m we $^o$C$^{-1}$ hr$^{-1}$ |
| | b$_{ice}$ | 4.0e-7 - 2.0e-6 | m$^3$ we W$^{-1}$ $^o$C$^{-1}$ hr$^{-1}$ |
| | b$_{snow/firn}$ | 4.0e-8 - 4.0e-7 | m$^3$ we W$^{-1}$ $^o$C$^{-1}$ hr$^{-1}$ |
| TIM$_3$ | a$_{ice}$ | 1.5e-4 - 3.0e-4 | m we $^o$C$^{-1}$ hr$^{-1}$ |
| | a$_{snow/firn}$ | 6.0e-5 - 2.0e-4 | m we $^o$C$^{-1}$ hr$^{-1}$ |
| | b$_{ice}$ | 1.0e-5 - 8.0e-5 | m$^3$ we W$^{-1}$ hr$^{-1}$ |
| | b$_{snow/firn}$ | 2.0e-7 - 4.0e-6 | m$^3$ we W$^{-1}$ hr$^{-1}$ |
| | p$_2$ | 0.01 - 0.4 | |
| ROR$_1$ | k | 1 - 30 | hr |
| | n | 1 - 5 | |
| ROR$_2$ | k$_{ice/soil}$ | 0.1 - 5 | hr |
| | k$_{snow/firn}$ | 20 - 100 | hr |
| | n$_{ice/soil}$ | 1 - 5 | |
| | n$_{ice/snow}$ | 1 - 5 | |
| ROR$_3$ | k$_{soil}$ | 0.1 - 5 | hr |
| | k$_{ice}$ | 0.1 - 5 | hr |
| | k$_{snow}$ | 10 - 50 | hr |
| | k$_{firn}$ | 50 - 300 | hr |
| | n$_{soil}$ | 1 - 5 | |
| | n$_{ice}$ | 1 - 5 | |
| | n$_{snow}$ | 1 - 5 | |
| | n$_{soil}$ | 1 - 5 | |





Gemma Coxon (University of Bristol) regarding the implementation of the limits of acceptability framework. Finally, we acknowledge the assistance given by Heiko Buxel (British Geological Survey) for collecting the on-ice temperature measurements. JDM, CRJ and JE publish with permission of the Executive Director of the British Geological Survey.



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
