# Peer review of "Glacio-hydrological melt and runoff modelling: a limits of acceptability framework for model selection"

_The Cryosphere, 2017_

## Referee Comment (RC1) · Anonymous Referee #1 · 19 Feb 2018

Review of Glacio-hydrological melt and runoff modelling: a limits of acceptability framework for model selection by J.D. Mackay et al.

This paper presents a truly comprehensive investigation into the performance of a set of glacio-hydrological models of varying complexity. Such models basically take climate and topographic data and calculate river discharge via model calculations of distributed glacier melt, linked to overland and river flow models. The models investigated here are all at what might judged as the simpler end of model configurations used within glaciology, as the melt models are variations on the well-established temperature index approach, and the runoff models are all reservoir models, again of varying complexity.

The key advance in the paper is the use of a comprehensive range of output measures used to judge model performance, and within that, the adoption of a sophisticated estimate of the limits of acceptability for each performance test based on uncertainties and errors within these datasets due to intrinsic errors or errors induced by sampling, spatial scale and the like. This is a powerful advance over most previous studies, which have used a limited set of performance measures, normally judged with rather simple statistics for model agreement, such as correlation or the various mean error statistics available. I think the argument made that such simplistic model/data comparison approaches have limited glaciological studies in comparison with other model disciplines, such as hydrology, is powerful, although it has been addressed occasionally within glaciology before. Rye et al. 2012, for instance, show how there is a need for multi-objective optimisation within melt modelling, as no single measure of model fit is adequate to fully capture the performance of any given model, and different outcomes, in terms of longer-term mass balance predictions are possible with equally acceptable models. The methodology used by Rye et al., and the overall scope of their investigation is different to the current study, but the over-arching aims seem very similar.

The paper sets out a clear methodology to develop the limits of acceptability for each 'test' dataset. As examples, the LOA as defined for ice melt include the need to allow that model-based estimates of melt when compared with in-situ stake ablation measurements can only be expected to be as close as the actual spread in melt over the equivalent node area within the model. This spread was calculated using high resolution terrestrial LIDAR scans made during the stake measurement campaign, and used to define 95% confidence bounds around the stake measurements. MODIS-based (and therefore 500m spatial resolution) snow cover estimates were transformed to use within the 50m resolution DEM of the study site with a monte-carlo approach in which for each MODIS snow pixel the relevant number of 50m snow pixels were distributed randomly 1000 times to generate confidence bounds around the snow cover-elevation measure used as a model test. Such processes were adopted for 33 data 'signatures'.

[Figure]

The comparison exercise was based on the use of three variations in temperature index model (of differing complexity), run with each of three runoff models, giving 9 model configurations. Each was then run with a randomised range of parameter values, yielding 45,000 calibration runs to be judged against each of the 33 'signatures'. This set of results is evaluated both in terms of measuring the various models' performance against the range of signatures, and also in terms of the signatures' discriminatory power. Ultimately, no single model nor set of parameters met all of the LOA across all the signatures, but the approach was capable of showing the 'trade-offs' within model configuration/parameterisation in terms of the different patterns of acceptability identified, and it also showed that additional complexity did not always lead to improved acceptability.

This might be seen as a limitation of the current study. It may be that such a comprehensive analysis of a range of models and data signatures would lead to a different 'specific' conclusion in terms of the best-fit model, or the most powerful discriminatory data, when applied to a different glacier system. I do feel, however, that whilst such a complex study as this one is not going to be possible for every model application, a more sophisticated methodology to evaluate model fit is important within glaciology. I think a key specific outcome of this study (which to my mind merits publication by itself) are the descriptions and methodologies use to generate the limits of acceptability for the range of data available. These will form a useful resource for future studies which may adopt this, or similar, methodologies. Additionally, as well as potentially improving the discriminatory power of models and our ability to discriminate between different model performance, work such as the current study, and a few earlier papers, will also allow a better understanding of the uncertainty inherent in model predictions. I think this is a careful and extremely thorough study, and I strongly recommend it for publication. I hope it gains impact and this type of more sophisticated evaluation of model performance gains traction within the glaciological community.

The paper is also commendably well written. I have very few specific corrections which

[Figure]

I feel should be made.

P2 L10 '. . .adhere to, . . .' (insert comma)

P3 L22 ' therefore we' (delete comma)

P3 L24 'definition imperfect' (delete comma)

P5 L15 change 'on' to 'of'

P9 L 25 delete 'an' before 'can'

P15 L1 Suggest rewording to 'As in may glaciated catchments, topography controls spatial temperature gradients to a large extent'.

P15 L7 Suggest change 'shallow' to 'reduce'

P18 L 6 'curves provide' (delete comma)

P24 Figure 8 caption or key needs to include explanation of the dots on Fig 8a

P40 L16 'all but two of' (delete both commas)

Reference Rye, C.J., Willis, I.C., Arnold, N.S. and Kohler, J., 2012. On the need for automated multiobjective optimization and uncertainty estimation of glacier mass balance models. Journal of Geophysical Research: Earth Surface, v. 117, doi:10.1029/2011JF002184

---

## Referee Comment (RC2) · D. Loibl (Referee) · 2 Mar 2018

The study presents an innovative approach to evaluate the performance of glacio-hydrological models. Basing on data from Virkisá river catchment in Iceland, the study demonstrates a framework to constrain the acceptability of results from different models against observational data. Different setups combining relatively simple glacier melt and discharge models are tested for their capability of reproducing measurement data results of melt, snow cover, and river runoff. Such a comprehensive framework for model evaluation is certainly an important contribution. I got the impression that the setup presented manuscript was designed very thoughtfully. The manuscript is also

very well written, with well-done figures illustrating many key aspects.

Nevertheless, I see three major weaknesses of the manuscript in its current form which need to be addressed before publication: (i) The weak precipitation data set, (ii) patchy result data, and (ii) that no scripts or technical details are presented.

(i) The AWS data used to force the models does not contain information on snowfall, and even rainfall with consecutive three above-freezing days was used. The manuscript provides few information of how much data is actually lost though this procedure (s. also ii), but I guess it is quite a lot in Iceland. These measures will certainly have substantial effects on glaciological-hydrological outputs and may account to some extent for the models inability to calculate winter melt. However, the authors use the data very thoughtful and have made quite some effort to ensure validity. Ideally, the framework should additionally be tested in a setting with high-quality AWS/snowfall data to constrain specific effects originating from input data characteristics. Arguably, this is beyond the scope of this manuscript; Nevertheless, I find it very important to discuss in more detail to what extent the strengths and weaknesses identified for individual model setups might also be affected by shortcomings in input/observation data.

(ii) Result data tends to be patchy, making it hard for the reader to get the broader picture. For instance, Fig. 13 only shows modeling results for May, Fig. 14 only the year 2013, Fig. 16 only selected months, etc. I totally understand why these selections were made. However, I strongly recommend to add full input and output datasets as supplements to show that the examples are no cherry picking, and to allow readers to see the greater picture.

(iii) Intuitively, I'd say the term 'framework' suggests that the aim is broad application, and I think the conclusions chapter encourages application and further testing of the LOA framework. However, the underlying structures and algorithms remain vague. If the aim is to let other scientists follow this approach, I suggest providing more detail on how to set up a LOA framework. As an open science and open source enthusiast, I

personally would like an open git repository where interested persons could access the code and maybe help developing it further; This would certainly boost the resonance to this good work.

In combination, (ii) and (iii) render it impossible to reproduce your analysis in the current state of the manuscript.

Ultimately, I find there is too much interpretation in the results chapter and suggest a clearer distinction between results and discussion.

Technical comments

Inconsistent order of citations P4L19 - Introduce, omit, or at least quote the abbreviation SEHR-ECHO P5L10 - Provide unit for slope (radians, I guess). Personally, I find slope angles in degree more convenient. Fig. 9: Why are boxes dark gray where the score is 0? Fig. 13: What is '1e3 m$^3$'?

---

## Referee Comment (RC3) · Anonymous Referee #3 · 6 Mar 2018

Summary of the manuscript This manuscript (ms) aims to presents "A limits of acceptability framework for model selection for glacio-hydrological melt and runoff modelling". For this purpose three model structures were calibrated and validated with ice volume change observations, satellite derived snow cover images and runoff data. The authors conclude that "it remains to be seen if the framework can be used", acknowledging that the results were "not necessarily more consistent across the full range of signatures". Further studies would be needed to investigate these conclusions.

Evaluation I think the topic of this manuscript is highly relevant and important for hydrological modelling and therefore it should be tackled thoroughly and with care. In recent

years many studies have compared model complexity and different observational data sets to investigate glacio-hydrological melt and runoff modelling. Accordingly, I would highly appreciate a concise and well-established framework of acceptability of glacio-hydrological models. However, I have my major doubts about the present ms and if it delivers what the title promises. My concerns are the following: i) the authors conclude that the results were "not necessarily more consistent across the full range of signatures" and that "it remains to be seen if the framework can be used". Based on the presented results I would agree with these conclusions and will provide comments how this can be tacked in a more thorough way below. But I also would argue that a framework that cannot provide consistent results and has to be investigated further is not publishable. ii) The evaluation metrics (equation 9) are based on user defined signatures thresholds, rather than well-established efficiency criteria. This makes it impossible for the reader to assess the performance of the model. iii) Inter-annual average altitude dependent simulated and observed snow cover is presented (Fig 12) rather than daily time steps. This makes it impossible to assess how the model performs during years with enhanced snowfall and reduced snowfall. Averages of inter-annual performances dilute the performance, making it impossible for the reader to assess the real performance. iv) Same argument is valid for runoff: in my opinion daily time-steps should be compared for a calibration and validation period. v) In my opinion, a validation of the modelling results based on the presented "user defined" evaluation metrics is inadequate, because it is not objective (If other thresholds were selected the results might have been completely different, making the presented analysis subjective). vi) Many of the concerns addressed above have been analyzed and discussed in recent works, some of which are referenced in this ms but not taken adequately into account. In my opinion the authors fail to connect to previous works and point out why this framework is novel and how this ms fills an existing knowledge gap.

I leave it up to the editors of "The Cryosphere" to decide if the present ms can be revised or should be resubmitted as a new ms. I would recommend to address the following concerns prior to publication:

Major concerns: 1) A framework has to work to be published. Accordingly, the framework should be able to reproduce adequately i) bi-annual glacier mass balances (accumulation and ablation phase) over several years, ii) daily snow cover ratio over several years to demonstrate that it works for snow intense and snow poor years. iii) daily runoff patterns. 2) If three model structures are used and they produce the same results, I would argue that the model structure is insensitive to the modelling approach. Nevertheless, the title claims to provide an acceptability framework for model structure. How can the framework say anything about model structure, it all the structures produce similar results? 3) Evaluation metrics should be consistent and comparable with previous literature. I understand that the chosen study site might lead to exceptionally low performance due to extreme weather patterns, but I would still use Nash-values for runoff, RMSE for glacier mass balances and a representative index for the snow cover area. This would enable a comparison of the model performance with previous works. 4) In my opinion, the ms can be significantly shortened, made more concise and the literature should be selected more carefully.

Specific recommendations: • Title: I find the title confusing and misleading. Please provide a more focused title. • Abstract: do models underpin the understanding of future climate change? I would argue that only the analysis of the results obtained with models can do that (it's a detail, but important, I recommend that the entire ms is revised and such flaw statements are corrected) • Introduction: in my opinion this chapter can be significantly shortened to a third of its current length. But I also think that it should be more targeted and the cited literature should be more focused. • Referencing: in my opinion 3 references are sufficient for one statement; in this ms up to 17 references are used of a single statement. This is misleading and not helpful for the reader. I think every references should be carefully chosen and only the most relevant references should be used (this would keep the reader focused on the topic of the ms). Also, I would recommend referencing the first publication that has investigated a topic, rather than referencing newer articles who have just build on previous works. Referencing is a tedious job, involves a lot of reading but should be taken seriously. • I would tend to disagree that this is the "first kind to apply a signature-based limits of acceptability (LOA) framework". In my opinion numerous studies have done this, simply in a different manner. • Figure 1: Why are meteo-data presented as an inlet in the map of the study site? Please stick to a coherent structure and provide all figures in a standard scientific style. • Observational data: in my opinion this chapter can also be significantly shortened. None of the data were collected by the authors, so the methods how they were collected can be found in the relevant literature. • Glacio-hydrological model: has the model never been published before? Does it have a name? Why not use a model that is well established? Then the description of the model could be shortened and the focus could be on the framework. • Driving climate data: The Icelandic Met Office (IMO) provides gridded reanalysis data (P, T and I). They are continuously updating their gridded data sets. Accordingly, I would recommend using the official data set of the IMO, rather than reproducing something that has been investigated for many years by the IMO. • Figure 3: It is impossible to assess how well the correction worked based on the presented data; see my previous comment. I recommend computing some relevant statistical values (see some of the cited literature). • Fig 4,5,67: please see my concern regarding aggregated monthly or inter-annual comparison of observed and simulated runoff and snow cover; What is the use of modelling daily time steps if only mean monthly results are evaluated? If the objective is only the get monthly means correctly than I recommend using a monthly time step in the modelling framework. • Fig 8: see my comment regarding the evaluation metrics; • Fig 9: I recommend making a table rather than a figure; • Fig 10: why select a case study which has only two glacier volume observations and 23 year of data gap between the two observations? There are numerous case studies which have bi-annual glacier mass balances, which would be a lot more valuable to establish a framework of acceptability for glacio-hydrological models; If the authors want to establish a transparent framework, I recommend selecting a case study with enough observational data to test the framework thoroughly. • Fig 11: in my opinion these results suggest that the framework does not work; none of the simulations are

acceptable, even according to the standards set by the authors;  c Fig 12: this figure illustrated the discrepancy between observed and simulated snow cover: In my opinion a acceptability framework should be able to identify if a model works during snow intensive and snow poor years. Here we only see that it never really works.

Final remark: I do think that this study can become an important contribution to hydrological modeling. However, all of the comments above would need to be accounted for and/or addressed. I would like to encourage the authors to have a thorough discussion on the purpose of this study. Furthermore, I encourage the authors to read some of the literature that is already referenced; many of the concerns addressed above have already been investigated and solutions have been presented. I would like to wish the authors lots of success and good luck with further research on such a framework.

---

## Author Response (AR1)

**tc-2017-268 response to referees (with revisions)**, **30th April 2018**.

Again, we would like to thank the referees and the editor for their feedback. We have now made our proposed revisions which we feel has resulted in a much improved manuscript. Below, we detail (in red) all of the revisions we have made to the manuscript in response to the referee comments. For complete transparency, we have kept all of the text from our final response document submitted on 6th April 2018 (FinalResponse_tc-2017-268.pdf) which includes the referee comments in black and our responses in blue. As well as our submission of the revised manuscript, we have also attached a mark-up document at the end of this document which details all of the revisions made. We look forward to the response from the editor at your earliest convenience.

We thank all of the referees and the editor for taking their time to read our manuscript and provide us with some very useful feedback. We received three reviews, two of which were overwhelmingly positive and one of which was largely critical of the manuscript; we greatly appreciate all of the comments. All referees clearly share our appreciation of the importance of this area of research, and as such, we hope our detailed responses below will enable us to contribute to this subject. We have taken care to address all comments, especially to those that require clarification or are critical for which we provide more detailed responses with clear justifications. We list below all of the referee comments in black followed by responses in blue. Any proposed revisions are highlighted with **bold** text. We look forward to the response from the editor at your earliest convenience.

**Anonymous Referee #1**

This paper presents a truly comprehensive investigation into the performance of a set of glacio-hydrological models of varying complexity. Such models basically take climate and topographic data and calculate river discharge via model calculations of distributed glacier melt, linked to overland and river flow models. The models investigated here are all at what might judged as the simpler end of model configurations used within glaciology, as the melt models are variations on the well-established temperature index approach, and the runoff models are all reservoir models, again of varying complexity. The key advance in the paper is the use of a comprehensive range of output measures used to judge model performance, and within that, the adoption of a sophisticated estimate of the limits of acceptability for each performance test based on uncertainties and errors within these datasets due to intrinsic errors or errors induced by sampling, spatial scale and the like. This is a powerful advance over most previous studies, which have used a limited set of performance measures, normally judged with rather simple statistics for model agreement, such as correlation or the various mean error statistics available. I think the argument made that such simplistic model/data comparison approaches have limited glaciological studies in comparison with other model disciplines, such as hydrology, is powerful, although it has been addressed occasionally within glaciology before. Rye et al. 2012, for instance, show how there is a need for multi-objective optimisation within melt modelling, as no single measure of model fit is adequate to fully capture the performance of any given model, and different outcomes, in terms of longer-term mass balance predictions are possible with equally acceptable models. The methodology used by Rye et al., and the overall scope of their investigation is different to the current study, but the over-arching aims seem very similar.

The paper sets out a clear methodology to develop the limits of acceptability for each 'test' dataset. As examples, the LOA as defined for ice melt include the need to allow that model-based estimates of melt when compared with in-situ stake ablation measurements can only be expected to be as close as the actual spread in melt over the equivalent node area within the model. This spread was calculated using high resolution terrestrial LIDAR scans made during the stake measurement

campaign, and used to define 95% confidence bounds around the stake measurements. MODIS-based (and therefore 500m spatial resolution) snow cover estimates were transformed to use within the 50m resolution DEM of the study site with a monte-carlo approach in which for each MODIS snow pixel the relevant number of 50m snow pixels were distributed randomly 1000 times to generate confidence bounds around the snow cover-elevation measure used as a model test. Such processes were adopted for 33 data 'signatures'.

The comparison exercise was based on the use of three variations in temperature index model (of differing complexity), run with each of three runoff models, giving 9 model configurations. Each was then run with a randomised range of parameter values, yielding 45,000 calibration runs to be judged against each of the 33 'signatures'. This set of results is evaluated both in terms of measuring the various models' performance against the range of signatures, and also in terms of the signatures' discriminatory power. Ultimately, no single model nor set of parameters met all of the LOA across all the signatures, but the approach was capable of showing the 'trade-offs' within model configuration/parameterisation in terms of the different patterns of acceptability identified, and it also showed that additional complexity did not always lead to improved acceptability.

This might be seen as a limitation of the current study. It may be that such a comprehensive analysis of a range of models and data signatures would lead to a different 'specific' conclusion in terms of the best-fit model, or the most powerful discriminatory data, when applied to a different glacier system. I do feel, however, that whilst such a complex study as this one is not going to be possible for every model application, a more sophisticated methodology to evaluate model fit is important within glaciology. I think a key specific outcome of this study (which to my mind merits publication by itself) are the descriptions and methodologies use to generate the limits of acceptability for the range of data available. These will form a useful resource for future studies which may adopt this, or similar, methodologies. Additionally, as well as potentially improving the discriminatory power of models and our ability to discriminate between different model performance, work such as the current study, and a few earlier papers, will also allow a better understanding of the uncertainty inherent in model predictions. I think this is a careful and extremely thorough study, and I strongly recommend it for publication. I hope it gains impact and this type of more sophisticated evaluation of model performance gains traction within the glaciological community.

The paper is also commendably well written. I have very few specific corrections which I feel should be made.

We greatly appreciate the positive and encouraging comments from the referee and echo the hope that this manuscript will stimulate further exploration of model evaluation approaches within the glaciology community and that it will provide a useful resource for those wishing to do so.

We agree that study of Rye et al. is somewhat different in overall aim to this study, but both were undoubtedly motivated by similar curiosities about model selection, model evaluation metrics and prediction uncertainty which warrants its inclusion in the introduction. **Accordingly, in our revised manuscript we will add additional text at the end of P4 L4 describing their findings.**

We have now added the following additional text (see P4 L10 in mark-up document) to the introduction which details the study of Rye et al. (2012) as an example of a multi-criterion approach to identify structural inadequacies in a distributed surface mass balance model:

"Rye et al. (2012) applied such an approach to 5 optimise a distributed surface mass balance model of two glaciers in Svalbard. They used ablation stake data to define three different features of the observations including mass balance at the stake locations, long term mass balance trend and mass

balance gradient. Using a multi-objective optimisation procedure, they identified structural inadequacies relating to how the mass balance gradient was simulated."

P2 L10 '. . .adhere to, . . .' (insert comma)

P3 L22 ' therefore we' (delete comma)

P3 L24 'definition imperfect' (delete comma)

P5 L15 change 'on' to 'of'

P9 L 25 delete 'an' before 'can'

P15 L1 Suggest rewording to 'As in may glaciated catchments, topography controls spatial temperature gradients to a large extent'.

P15 L7 Suggest change 'shallow' to 'reduce'

P18 L 6 'curves provide' (delete comma)

P24 Figure 8 caption or key needs to include explanation of the dots on Fig 8a P40

L16 'all but two of' (delete both commas)

Thank you for bringing these issues to our attention. **All of the above edits will be addressed in the revised manuscript.**

All of these edits have been undertaken as suggested.

Reference Rye, C.J., Willis, I.C., Arnold, N.S. and Kohler, J., 2012. On the need for automated multiobjective optimization and uncertainty estimation of glacier mass balance models. Journal of Geophysical Research: Earth Surface, v. 117, doi:10.1029/2011JF002184

**David Loibl (Referee #2)**

The study presents an innovative approach to evaluate the performance of glaciohydrological models. Basing on data from Virkisá river catchment in Iceland, the study demonstrates a framework to constrain the acceptability of results from different models against observational data. Different setups combining relatively simple glacier melt and discharge models are tested for their capability of reproducing measurement data results of melt, snow cover, and river runoff. Such a comprehensive framework for model evaluation is certainly an important contribution. I got the impression that the setup presented manuscript was designed very thoughtfully. The manuscript is also very well written, with well-done figures illustrating many key aspects.

Nevertheless, I see three major weaknesses of the manuscript in its current form which need to be addressed before publication: (i) The weak precipitation data set, (ii) patchy result data, and (ii) that no scripts or technical details are presented.

We appreciate these overall very positive comments from the referee, particularly with regards to the quality of the writing and presentation, as also noted by Anonymous Referee #1 and the Editor, of which we did indeed spend considerable effort in presenting in as clear and transparent way as possible. We also appreciate the clarity with which the referee has detailed their suggested weaknesses for improvement of the manuscript. We will respond to these comments sequentially as laid out by the referee.

(i) The AWS data used to force the models does not contain information on snowfall, and even rainfall with consecutive three above-freezing days was used. The manuscript provides few information of how much data is actually lost though this procedure (s. also ii), but I guess it is quite a lot in Iceland. These measures will certainly have substantial effects on glaciological-hydrological outputs and may account to some extent for the models inability to calculate winter melt. However, the authors use the data very thoughtful and have made quite some effort to ensure validity. Ideally, the framework should additionally be tested in a setting with high-quality AWS/snowfall data to constrain specific effects originating from input data characteristics. Arguably, this is beyond the scope of this manuscript; Nevertheless, I find it very important to discuss in more detail to what extent the strengths and weaknesses identified for individual model setups might also be affected by shortcomings in input/observation data.

We completely agree that the input data has a major control on model simulations and it is therefore important to stress potential weaknesses that could arise in the simulations as a result of deficiencies in the driving data. Although we did hint at these deficiencies in our discussion (P38 L5), particularly in relation to the ice melt and snow distribution signatures, we agree that we should have provided a more comprehensive discussion of this, particularly in relation to the river discharge simulations which are undoubtedly also susceptible to errors induced by deficiencies in precipitation data. **Accordingly, for the revised manuscript we will provide additional discussion of this aspect.**

**In conjunction, we intend to provide additional analyses of the bias-correction procedure used for precipitation (in response to comments from Anonymous Referee #3), which will include details of lost data due to freezing days. Accordingly, the additional discussion points will draw on the results from these analyses so that weaknesses in model simulations can be better related to weaknesses in the driving data.**

We have now added considerable text to the discussion to emphasise the importance of the driving climate data, particularly in relation to precipitation for which we don't have any snowfall data (see P39 L17 onwards in mark-up document).  In particular, we emphasise that:

- "there were fewer observations during winter months and none at all before 2009"
- "while the bias-corrected precipitation time-series was well correlated over a three-day time-step, it was not at an hourly time-step"
- "precipitation observations were all collected at the bottom of the catchment and therefore driving precipitation data at the top of the catchment are less certain"

In the previously submitted manuscript, we already discussed how these errors could explain some of the simulated biases for the ice and snow signatures. We have now added to this with an additional example of how these errors could further propagate to the river discharge signatures:

"Furthermore, given the strong coupling between snow, ice and river runoff, deficiencies in capturing the snow and ice signatures could also propagate through the hydrological representation of the catchment. For example, one could imagine how errors in the spatial distribution of snow could perturb the timing of runoff through the catchment given 20 that snow distribution influences the behaviour of the semi-distributed runoff-routing routine employed in the GHM. Such perturbations are likely to impact the ability of the GHM to capture the full range of river discharge signatures."

We finish, by reiterating the importance of the driving climate data:

"Accordingly, it is important to stress the influence that biases in the driving climate data could have on the model acceptability across the different signatures."

We also noted that in section 2.4.1 of the original manuscript, we speculated that, "poor replication of the timing of hourly rainfall events should not influence the GHM's ability to capture the river discharge signatures". While we still feel this is true for shorter-timescale signatures given that we have complete rainfall data for 2013 and 2014, the above statement about the propagation of error from the ice and snow representation in the GHM through to the river discharge simulations somewhat contradicts this. As such, we've reworded this statement to read: "…poor replication of the timing of hourly rainfall events should have minimal influence on the GHM's ability to capture the river discharge signatures" (P14 L5 in mark-up document).

(ii) Result data tends to be patchy, making it hard for the reader to get the broader picture. For instance, Fig. 13 only shows modelling results for May, Fig. 14 only the year 2013, Fig. 16 only selected months, etc. I totally understand why these selections were made. However, I strongly recommend to add full input and output datasets as supplements to show that the examples are no cherry picking, and to allow readers to see the greater picture.

Yes we understand your concern here. To be clear, we made these selections rather than displaying the full time-series for two main reasons: i) it helps to focus the reader on those aspects of the hydrograph being analysed in the text; and ii) differences in model behaviour are more difficult to visualise when the simulation data are compressed in the graph due to displaying the full time-series. Please also note that these selections were made because they provide ideal periods to analyse different aspects of model behaviour. For example, May 2013 was chosen for Figure 13 because it is a period with almost no rainfall at the start of the melt season, and so differences in the river discharge simulations can be considered a function of the melt behaviour of the model rather than the rainfall-runoff behaviour. We could have chosen 2014 (see similar enhanced melt at start of melt season in the figure included below), but because of the presence of a number of rainfall events, we used 2013. In contrast, Figure 16 include periods where melt is insignificant compared to rainfall, and which include a range of peak river discharge magnitudes. Nevertheless, we completely agree that, for complete transparency, we should include the full input/output time-series for these simulations. **Accordingly, as part of the revised manuscript we will include an additional section in the appendix with these figures.**

We have now included these additional figures in the Appendix (see P51&52 of the mark-up document). These include the watershed total precipitation, average temperature and incident solar radiation data used to drive the models over the complete observed river discharge observation period. We have also included the complete simulations of snow melt, ice melt and river discharge (plotted with the observations) using the different GHM configurations.

Additionally, we make the reader aware of these figures immediately after referencing Figure 13 in the main body of the text (P30 L7 in the mark-up document).

(iii) Intuitively, I'd say the term 'framework' suggests that the aim is broad application, and I think the conclusions chapter encourages application and further testing of the LOA framework. However, the underlying structures and algorithms remain vague. If the aim is to let other scientists follow this approach, I suggest providing more detail on how to set up a LOA framework. As an open science and open source enthusiast, I personally would like an open git repository where interested persons could access the code and maybe help developing it further; This would certainly boost the

resonance to this good work. In combination, (ii) and (iii) render it impossible to reproduce your analysis in the current state of the manuscript.

We certainly share your enthusiasm for being open with data, publications and computer code, and given that this work is funded by a Natural Environment Research Council (NERC) studentship, all aspects of this work come under the NERC data policy (see here: http://www.nerc.ac.uk/research/sites/data/policy/) which requires all data to become open within 2 years and code to be preserved for others to use. We agree that git would be an ideal repository, however we are mindful that the code written for this study is completely tailored for our study basin and its data format. To produce a usable set of open-source scripts of the code used to calculate the LOA around the river discharge signatures (for example) would take some considerable effort. That is not to say that this is something we won't do in the future, just that if/when we do this, we'd like to do it properly and arguably, this would warrant a publication in itself. We are also surprised that you feel the underlying structure and algorithms remain vague. We feel we have been as open as we can be about the methods used. For example:

- GHM: All model equations are well established and either referenced or written in full (P9 L11 onwards).
- Driving data: all data sources have been referenced where possible, and any bias-correction procedures have been referenced and/or detailed (sections 2.2 and 2.4).
- Signatures and limits of acceptability: All signatures are listed in Table 1 and referenced in the text (e.g. P19 L18) All derived LOA are also shown in Table 1 and methods to derive these are either referenced or explained in the text (section 2.5).

Of course, as with any publication, we would also be happy to receive correspondence from other researchers who may have further questions about the methods used for which we would provide advice and share code where possible. **We feel it is important to make this clear to the reader and as such, in our revised manuscript we will include some text at the end with regards to sharing of the code.**

We have now included an additional 'code and data availability' statement to make this clear to readers (see bottom of P42 in mark-up document) which reads:

"For persons interested in applying a similar signature-based LOA approach for model evaluation, we would encourage them to contact the authorship who are open to providing advice and sharing data and code where possible."

Ultimately, I find there is too much interpretation in the results chapter and suggest a clearer distinction between results and discussion

We purposely included some interpretation in the results section to help guide the reader through what are a complex set of results. However, we appreciate your preference for a complete separation of the results and discussion sections. **On re-reading the results, we've identified 10 portions of text with interpretation that can be reworded and moved into the discussion section: P23 L17, P25 L7, P29 L8, P30 L12, P31 L9, P32 L5, P32 L14, P32 L34, P35 L9, P35 L33. Accordingly, our revised manuscript will include these modifications.**

We have completely removed eight of the ten identified portions of text from the document as they made statements which were already included in the discussion section. The remaining two statements have been reworded and included in the discussion (P40 L1 and P40 L9 in the mark-up document).

Inconsistent order of citations P4L19

We used the Copernicus Latex template and we assumed these would be correctly ordered before publication, but **we are happy to re-order the citations ourselves if the editor requires.**

We have gone through the entire manuscript and re-ordered them manually (first by date, then by author).

Introduce, omit, or at least quote the abbreviation SEHR-ECHO P5L10

**This will be addressed in the revised manuscript.**

Removed as suggested (see P4 L27 in mark-up document).

Provide unit for slope (radians, I guess) Personally, I find slope angles in degree more convenient.

**This will be addressed in the revised manuscript.**

These have now been included (see P17 in mark-up document). Note, the units are in $m^3\ s^{-1}$ per section of the flow duration curve. See in the attached figure of the FDC (white dotted line is the mean FDC and blue bars indicate uncertainty). The volume under the curve is equivalent to the integral between two exceedance limits on the x-axis. The slope, is the change in discharge per flow exceedance section. We've made this clear in the table caption in the revised manuscript.

[Figure]

Fig. 9: Why are boxes dark gray where the score is 0?

This is rounding issue in the code used to generate these plots – i.e. those scores were not equal to zero, but very close. Indeed, this is also an issue for Figure 15 which we hadn't noticed. **These will be corrected in the revised manuscript.**

We have now modified these figures so that non-zero scores that round to zero for the plot are accompanied by a plus or negative sign to indicate the sign of the score, and to indicate that they are indeed non-zero and therefore unacceptable. Furthermore, additional text has been added to the figure captions to make clear to the reader that these scores are non-zero (see Figure 9, P27 and Figure 15, P33 in the mark-up document).

Fig. 13: What is '1e3 m3'?

This is accepted standard exponential notation, which in this example is equivalent to $1 \times 10^{-3}$ m$^3$. It was used to avoid excessively large numbers on the y-axis.

**Anonymous Referee #3**

This manuscript (ms) aims to presents "A limits of acceptability framework for model selection for glacio-hydrological melt and runoff modelling". For this purpose three model structures were calibrated and validated with ice volume change observations, satellite derived snow cover images and runoff data. The authors conclude that "it remains to be seen if the framework can be used", acknowledging that the results were "not necessarily more consistent across the full range of signatures". Further studies would be needed to investigate these conclusions.

I think the topic of this manuscript is highly relevant and important for hydrological modelling and therefore it should be tackled thoroughly and with care. In recent years many studies have compared model complexity and different observational data sets to investigate glacio-hydrological melt and runoff modelling. Accordingly, I would highly appreciate a concise and well-established framework of acceptability of glaciohydrological models. However, I have my major doubts about the present ms and if it delivers what the title promises. My concerns are the following: i) the authors conclude that the results were "not necessarily more consistent across the full range of signatures" and that "it remains to be seen if the framework can be used". Based on the presented results I would agree with these conclusions and will provide comments how this can be tacked in a more thorough way below. But I also would argue that a framework that cannot provide consistent results and has to be investigated further is not publishable. ii) The evaluation metrics (equation 9) are based on user defined signatures thresholds, rather than well-established efficiency criteria. This makes it impossible for the reader to assess the performance of the model. iii) Inter-annual average altitude dependent simulated and observed snow cover is presented (Fig 12) rather than daily time steps. This makes it impossible to assess how the model performs during years with enhanced snowfall and reduced snowfall. Averages of inter-annual performances dilute the performance, making it impossible for the reader to assess the real performance. iv) Same argument is valid for runoff: in my opinion daily time-steps should be compared for a calibration and validation period. v) In my opinion, a validation of the modelling results based on the presented "user defined" evaluation metrics is inadequate, because it is not objective (If other thresholds were selected the results might have been completely different, making the presented analysis subjective). vi) Many of the concerns addressed above have been analyzed and discussed in recent works, some of which are referenced in this ms but not taken adequately into account. In my opinion the authors fail to connect to previous works and point out why this framework is novel and how this ms fills an existing knowledge gap.

We echo the initial thoughts by Anonymous Referee #3 on the relevance of this kind of research and the need for these aspects to be tackled thoroughly and with care. This is certainly something we aimed to do in undertaking this work. The points above provide a useful overview of the referee's six

primary concerns and critiques of the manuscript, the majority of which are covered by subsequent, more specific comments below. As such, most of our detailed responses will be outlined later under these specific comments, but we feel it would also be useful to provide some initial thoughts before getting into these details.

On reading the full set of comments from Anonymous Referee #3, it is clear that one issue they have with the manuscript is that we are not using what could be termed 'traditional' means to evaluate the error, or skill, of the different models. For example, a number of critiques point to the fact that we are not evaluating 'daily' runoff or snow coverage patterns (although this is not entirely accurate, e.g. see Figures 13 and 16 for sub-daily runoff patterns) and that we are not using 'well-established efficiency criteria'. While we understand the referee's preference to use these kinds of criteria for evaluating model efficiency, we believe we set out clearly that this was not the premise of this manuscript. Its purpose, as outlined by the aims in the introduction (P4 L30), is to test a signature-based LOA approach which has been applied in hydrology (for good reason – see comments below on traditional efficiency criteria), and which we believe could also benefit the glaciology community given the similar problems of model equifinality (e.g. for melt models) demonstrated in the literature. Our thesis, which aligns with the recent literature (e.g. P4L9), is that we must go beyond the use of error criteria such as the RMSE or NSE (which describe an averaged or aggregated model skill, biased to high observational values) if we are going to diagnose deficiencies in model process representation, and support the identification of better model structural components, very importantly taking into account observational data uncertainty.

On reflection, we appreciate that the title of manuscript does not completely reflect the aims of the study and could be read as suggesting that we are presenting a definitive model selection framework when in actual fact the point of the study is to test the framework for identifying model deficiencies and selecting acceptable models. We feel at least some of the critique could stem from this initial framing of the study. **Indeed, as outlined in later comments, we suggest a small re-wording of the title in the revised manuscript that would help to frame this study properly**.

At this point, we also feel it would be worthwhile responding to the criticism about the subjectivity of the LOA (point v above) which is not explicitly raised in the remaining referee comments. To be clear, our use of the word objective is referring specifically to the definition of the acceptability thresholds and not the signatures themselves. **We perhaps have not made this completely clear, and if so will revise the text to ensure this is.** Of course, the choice of signatures will not be the same from study region to study region. For one, it will depend on what data are available. For example, we used river discharge signatures that characterise sub-daily behaviour of the system. Where only daily data are available, this won't be possible. The choice of signatures will also depend on which processes one is most interested in interrogating. For example, someone testing a distributed, physically based, snow-redistribution model who has high resolution snow depth measurements may derive signatures that thoroughly interrogate these detailed aspects of the model behaviour. For our study, we have derived a set of signatures that broadly characterise the glacio-hydrological behaviour of the basin based on the data we had available. However, these are by no means absolute. On the contrary, we would encourage others to experiment with different signatures to further interrogate model acceptability across different regions. Regardless of the choice of signatures, the LOA for a given signature should be defined from available information on observation data uncertainty. This definition of the acceptability criteria is the objectivity we refer to which is far more objective than some arbitrary efficiency criteria (see later responses).

To make this clear to the reader we have included some additional text at the end of the discussion section to emphasise that the choice of signatures is subject-specific (see P41 L31 of the mark-up document). It reads:

"Certainly, it's important to emphasise that these future applications need not adopt the same 33 signatures used in this study. On the contrary, the choice of signatures will always depend somewhat on the availability of data at a given study site as well as the complexity (e.g. spatio-temporal resolution) of the model(s) being interrogated. Indeed, future users should be encouraged to experiment with different signatures (where data permits) particularly if they wish to focus on other process representations within their GHM. Study sites with good observation data and understanding of data uncertainty would be ideal candidates for these future applications."

Additionally, we noted that in the original abstract, discussion and conclusions, we frequently referred to 'the LOA framework' which, on reflection we feel comes across as though we are proposing a definitive application of a LOA framework. As noted above, this is not the case, and accordingly we have changed a number of these instances to read 'a LOA framework' which we feel better reflects the fact we are presenting an application of a LOA framework (e.g. see P1 L18 in the mark-up document).

Finally, before proceeding to individual responses, we feel it is important to clarify the meaning of the two quotations from the manuscript which have been used in the above comments, but which we consider have been taken out of context and therefore used somewhat unfairly within the critique of the manuscript:

the authors conclude that the results were "not necessarily more consistent across the full range of signatures" and that "it remains to be seen if the framework can be used"

Regarding: "not necessarily more consistent across the full range of signatures"

This is referring to model complexity and has nothing to do with the suitability of the framework. To be clear, the full quotation from the manuscript reads:

"When evaluated against individual signatures, the more complex model formulations did improve model simulations in some cases. However, they were not necessarily more consistent across the full range of signatures, emphasising the need to exercise caution and properly evaluate if additional complexities are justified."

We believe this is an important point to emphasise, particularly in relation to representing water flow pathways through glaciated catchments. As we discuss in the introduction, linear reservoirs are widely used in glaciated regions to represent water flow pathways, but the form that these configurations should take is primarily down to a particular modellers own perceptions, experience in the field, model code affiliation etc…and rarely based on any formal exploration of model appropriateness. Note, we reference the paper of Hannah (2001) as one exception to this. One might expect (as we did), that given the known non-linearity in runoff behaviour in glaciated catchments because of the unique storage behaviour of snow and ice (e.g. see Jansson 2003), that a more complex (more non-linear) routing structure like $ROR_3$ in our study would better capture river flow regime over seasonal to sub-daily timescales – but we clearly show, using the range of river discharge signatures, that that isn't the case.

Regarding: "it remains to be seen if the framework can be used"

The full quotation from the manuscript reads:

"While all, but two, of the signatures demonstrated discrimination power, none of the 45,000 different model compositions tested in this study were able to capture them within their LOA simultaneously. Therefore, it remains to be seen if the framework can be used in this way, although we suggest that applications that go beyond examining the melt and runoff-routing structural uncertainties may prove more fruitful in obtaining a behavioural population of models."

So, to be clear we are reporting our finding for the second aim of the study as outlined in the introduction (P4 L29) which was to test the LOA framework for its ability to:

"ii) constrain a prior population of model structures and parameterisations down to a smaller population of acceptable models, indicating the framework's usefulness for reducing prediction uncertainty."

This is not to say it remains to be seen if the framework can be use as we know it can be used from applications in hydrology as outlined in the introduction (P4 L15). It is to say that finding a set of acceptable model structure/parameterisations could prove to be more difficult than we first thought. We should embrace this as something to work on (e.g. through the development of more suitable models) not something to dismiss.

Major concerns: 1) A framework has to work to be published. Accordingly, the framework should be able to reproduce adequately i) bi-annual glacier mass balances (accumulation and ablation phase) over several years, ii) daily snow cover ratio over several years to demonstrate that it works for snow intense and snow poor years. iii) daily runoff patterns.

A framework has to work to be published: We feel there might be some confusion as to what the purpose of this framework is and we expect this is linked to issues with the title. To be clear, it is not up to the framework to reproduce the observations. It is up to the models (including prescribed boundary conditions). The framework is there simply to indicate if a given model composition (i.e. structure and parameter set) is acceptable or not i.e. to determine if it is able to capture the signatures within their observation uncertainty. If none of the models are able to do this, this does not imply that the framework doesn't work, it implies that the models are not acceptable.

Accordingly, the framework should be able to reproduce adequately i) bi-annual glacier mass balances (accumulation and ablation phase) over several years: Again, it's important to reiterate here that it is not up to framework to reproduce available observation data. The framework's purpose is to tell the user whether a model is acceptable or not and what aspects of the system it succeeds or fails to capture (as defined by the signatures). In our case, we only have one set of melt season and one set of winter season ablation measurements (we'll discuss data availability below in a different comment). For other studies that have multiple years of ablation stake data, these could also be incorporated into the framework by adopting additional signatures as long as the uncertainty can be quantified, a point which we make in the discussions/conclusions.

ii) daily snow cover ratio over several years to demonstrate that it works for snow intense and snow poor years. iii) daily runoff patterns: Yes, as we noted above, we sympathise with the referee's preference for the more traditional analysis of model performance (e.g. through comparing time-series and using efficiency criteria). However, we must stress that this quite simply is not the point of this manuscript. We clearly outline that we are applying a different type of model evaluation technique: one that is based on signatures rather than time-series/efficiency criteria, which allows one to account for evaluation data uncertainty when making decisions about model appropriateness.

With regards to snow cover, it would have been possible to increase the number of snow signatures adopted in the study to incorporate snow intense/snow poor years. Doing this on a daily scale would have been difficult, mainly because good data are sparse (as noted in the methodology) and even when data do pass the QA, they often only cover parts of the catchment (presumably due to high relief and cloud cover). Accordingly, we deemed some degree of aggregation necessary to average out these discrepancies. In conjunction with this, we were also mindful that we already had 33 signatures to discuss, and given that it was clear none of the tested model compositions could capture the seasonal distribution of snowfall, we did not deem it necessary to further interrogate the models on extremes as the extra analysis, figures and text required simply was not worth it.

With regards to river discharge, we do relate model deficiencies identified in the framework to time-series of observed and simulated hourly river discharge for different flow regimes (Figures 13 and 16). In fact, given that river discharge signatures are well established in the literature (and because that's the thing that ultimately impacts downstream communities), we use a wide range of metrics over a range of timescales from monthly (monthly mean flows) to hourly (Integral time and peak flow hour – see Table 1).

2) If three model structures are used and they produce the same results, I would argue that the model structure is insensitive to the modelling approach. Nevertheless, the title claims to provide an acceptability framework for model structure. How can the framework say anything about model structure, it all the structures produce similar results?

Indeed, we found that all of the tested model structures were deemed unacceptable, i.e. they exhibit deficiencies that go beyond the known uncertainties of the evaluation data, and therefore none met the criteria to be 'selected'. On reflection we agree that we could have worded the title better so that it's clear we are testing the framework, rather than presenting it as a definitive model selection tool. Furthermore, we evaluated the framework for its ability to identify model deficiencies as well as for model selection which again, is not reflected in the title. As a reminder, and as set out at P4 L29, the aims of the study were to investigate the LOA framework utility for:

1) Diagnosing deficiencies in different model structures.
2) Constraining a prior population of model structures and parameterisations down to a smaller population of acceptable models.

Accordingly, we propose renaming the manuscript to better reflect both of these aims and emphasise the fact that we are applying the LOA framework for this purpose: **'Glacio-hydrological melt and runoff modelling: application of a limits of acceptability framework for model comparison and selection'.** We feel this is much more in line with the aims of the manuscript.

We have renamed the revised manuscript as described (see P1 of mark-up document).

3) Evaluation metrics should be consistent and comparable with previous literature. I understand that the chosen study site might lead to exceptionally low performance due to extreme weather patterns, but I would still use Nash-values for runoff, RMSE for glacier mass balances and a representative index for the snow cover area. This would enable a comparison of the model performance with previous works.

We're not completely sure if this is: i) a request to include these metrics on top what we have already presented; or ii) a critique of using signatures instead of these metrics (i.e. a critique of the framework). Accordingly, we will provide two responses here:

**i) We can include an additional table of these evaluation metrics in the appendix of the revised manuscript purely for comparison to other studies** (see later response for details of this). We chose not to do this, simply because the manuscript is already quite large, and these metrics have little to do with the framework. However, if the editor deems this necessary, we will include them.

ii) The use of signatures instead of well-established efficiency criteria underpins the methodology of the LOA framework. We do feel we've clearly explained and justified this in the introduction, but to be clear we'll summarise below exactly why signatures have been adopted in past applications instead of efficiency criteria:

1) What do efficiency criteria tell you about model efficiency? A key issue with using efficiency criteria to evaluate models is that they tell you very little about where the model is going wrong – they are an average of the model performance over a given observation dataset. We would ask, what does an NSE of 0.7 against observed river-flow time-series actually mean? What does it tell you about the model's ability to represent slow-release flow flows? What does it tell you about the model's ability to emulate the responsiveness (flashiness) of the system? Furthermore, all of these types of metrics are biased towards certain aspects of observation data. For example, it's been long recognised that the RMSE and NSE, both widely used to evaluate fit to river flow time-series, are biased towards fitting the highest flows (e.g. see Gupta et al 1998). MAPE on the other hand biases low flows. So using these not only provides very little information on where the model is working and where it is not, they also provide an overall biased indication of model efficiency.

2) Use of global efficiency criteria exacerbates the equifinality problem: Given 1), it's clear that evaluating models based on global metrics of model fit will give rise to multiple model structures and parameterisations that produce similar overall model fit leading to higher prediction uncertainty (e.g. see Gupta et al 2008 referenced in manuscript).

3) How do you define acceptability criteria of a model efficiency metric? The advantage of using signatures as evaluation metrics, is not only that they allow you to analyse specific aspects of a model's behaviour, but that quantitative LOA can be defined around these based on information about observation uncertainty providing the ability to evaluate model performance within the uncertainty (see P4 L9) of the evaluation metrics (especially important in mountain regions where observation data are often riddled with uncertainty). Acceptability criteria based on efficiency metrics have been used widely in the past. E.g. for river flow predictions, NSE > 0.6 is frequently used. However, this number is completely subjective (some use NSE > 0.5). What's more, given 1), one has no idea about what kind of model errors are being introduced to the predictions based on these criteria.

We appreciate that using signatures as the basis for model evaluation may seem controversial, but as we clearly set out in the introduction, it is something that has been done in the hydrology community and something that we feel is completely relevant to the glaciology community as well.

4) In my opinion, the ms can be significantly shortened, made more concise and the literature should be selected more carefully.

See Reponses to specific comments about text length and literature.

Specific recommendations: Title: I find the title confusing and misleading. Please ´ provide a more focused title

We agree that the title could be more focussed **and we have proposed modifying it for the revised manuscript.**

Abstract: do models underpin the understanding of future climate change? I would argue that only the analysis of the results obtained with models can do that (it's a detail, but important, I recommend that the entire ms is revised and such flaw statements are corrected)

We agree, it's a technicality, but an important one. Thank you for pointing this out. **We will change this and any similar statements so that it reflects the fact that it is the analysis of the projections from models that underpin our understanding.**

We identified two uses of this statement and have changed both to read "Glacio-hydrological models (GHMs) allow us to develop an understanding of…" (see P1 L1 and P1 L20 in mark-up document).

Introduction: in my opinion this chapter can be significantly shortened to a third of its current length. But I also think that it should be more targeted and the cited literature should be more focused.

We appreciate the introduction is longer than average, although we really can't see how we could remove two-thirds of it. To do so would surely mean removing key elements of the literature review. If this is the referee's recommendation, could we ask them to be more specific about what aspects of the introduction they believe should be removed?

We are also unsure as to what exactly 'more targeted' means. We presume this could be related to point vi from their initial comments which reads: "Many of the concerns addressed above have been analysed and discussed in recent works, some of which are referenced in this ms but not taken adequately into account. In my opinion the authors fail to connect to previous works and point out why this framework is novel and how this ms fills an existing knowledge gap."

Again, we are struggling to understand how to address this. We feel we've been explicit about what has and hasn't been done before. If the referee thinks we have missed something, could they please specify what this is? Which referenced material has not been taken into account properly? Which aspect of the novelty is not made clear? We do hope they can provide this information so that we can revise the introduction if required.

Referencing: in my opinion 3 references are sufficient for one statement; in this ms up to 17 references are used of a single statement. This is misleading and not helpful for the reader. I think every references should be carefully chosen and only the most relevant references should be used (this would keep the reader focused on the topic of the ms).

On reflection, we agree with the referee that the 17 citations (P19 L18) could be revised down to a smaller number. All of these were included simply because all were used to help determine which signatures to use in this study. However, several of the more recent publications include the majority of these and as such **we will edit this for the revised manuscript.**

We have revised this down to four publications which include the majority of the signatures used in this study (see P21 L5 in mark-up document).

We also acknowledge that the four citations used on P9 L7 could be revised down to the Huss *et al.* (2010) citation only, given that it is this study that best demonstrates that the delta-h parameterisation exhibits behaviour comparable to complex 3-D finite-element ice flow models. The other cited studies demonstrate additional applications of the approach rather than specifically testing it against complex ice-flow models. Accordingly, **we will edit this for the revised manuscript.**

This has been edited as described in the revised manuscript (P9 L17 in mark-up document).

While we appreciate the referee's preference for including a maximum of three citations, we do feel that in some cases it is useful to include more to provide the reader with some appreciation of the breadth of research that has been done a particular subject. For example, we use five references on P1, L20: 'Computational GHMs underpin our current understanding of how future climate change will affect river flow regimes in glaciated watersheds (Ragettli et al., 2016; Singh et al., 2016; Teutschbein et al., 2015; Lutz et al., 2014; Radic and Hock, 2014).' We chose these five studies because they cover applications of GHMs across a number of different regions (e.g. Lutz et al., 2014 in Asia vs Teutschbein et al., 2015 in Northern Europe) and at a number of different scales (e.g. Radic and Hock, 2014 global scale vs Ragettli et al., 2016 catchment scale).

Similarly, on P1 L22, we use six references: 'A variety of GHM codes exist (e.g. Bergström, 1997; Ciarapica and Todini, 2002; Schulla, 2015; Huss et al., 2008b; Boscarello et al., 2014; Schaefli et al., 2014), each of which include...' Here we chose a handful of the most well established GHM codes and some that are less well established but which demonstrate different levels of model complexity (e.g. the semi-distributed HBV model of Bergström, 1997 vs the physically-based TOPKAPI model of Ciarapica and Todini, 2002). WaSiM is another widely used model code (Schulla, 2015) while the model of Huss et al., 2008 has also been widely applied within a glacio-hydrological context. The model of Schaefli et al. (2014) is a recent addition to available GHMs and is unique in that it was originally developed for eco-hydrological modelling, but has been applied for glacio-hydrological modelling as well.

Also, I would recommend referencing the first publication that has investigated a topic, rather than referencing newer articles who have just build on previous works. Referencing is a tedious job, involves a lot of reading but should be taken seriously.

We understand the referee's preference for citing the oldest publications on a given topic. We spent considerable time selecting the most relevant citations to include in the introduction (although we appreciate that on the two occasions noted above we could have been more prudent). Our overwhelming feeling was that for the topics covered, the most relevant literature was generally also relatively recent. This was not surprising, given the relative novelty of signature-based applications of the 'limits of acceptability' methodology and the clear need for new model interrogation/selection strategies within the field of glaciology. However, we would be open to suggestions of specific citations that the referee feels have been missed and would add important additional points to the introduction.

I would tend to disagree that this is the "first kind to apply a signature-based ´ limits of acceptability (LOA) framework". In my opinion numerous studies have done this, simply in a different manner.

We agree that this is not the first of its kind to apply a signature-based limits of acceptability (LOA) framework, but do not claim it to be. We consider that we clearly reviewed past studies that have applied a signature-based limits of acceptability framework (P4 L11). Rather, we state that: "This study is the first of its kind to apply a signature-based LOA framework for a multi-GHM-structure evaluation." i.e. we have taken something that has been used in hydrology and applied it in a glacio-hydrological context. We feel this is quite clear.

Figure 1: Why are meteo-data presented as an inlet in the map of the study site? Please stick to a coherent structure and provide all figures in a standard scientific style.

The meteo-data are presented as an *inset* because it is part of the catchment description. We also use insets in Figure 10. We do feel this is in keeping with standard scientific style and makes for a more compact figure so we would prefer to keep it as it is.

Observational data: in my opinion this chapter can also be significantly shortened. None of the data were collected by the authors, so the methods how they were collected can be found in the relevant literature.

We agree there is scope for shortening this section. **We will do so in the revised manuscript.**

We have now removed a significant portion of the text describing the climate data in section 2.2.1. In particular, we have removed the text describing the rain-gauge apparatus (P7 L14 in mark-up document) which on reflection, we feel was providing too much detail. Also, we've removed the description of how the ICRA precipitation data was produced (P7 L24 in mark-up document) given that this can be found in the cited literature.

Additionally, we have also removed the first sentence describing the snow coverage data in section 2.2.3 which only serves to reiterate a point covered in the introduction (P8 L11 in mark-up document).

Glacio-hydrological model: has the model never been published before? Does it have a name? Why not use a model that is well established? Then the description of the model could be shortened and the focus could be on the framework.

No the model code has not been published and it doesn't have a name. The reason we didn't use someone else's model code was essentially because there wasn't one out there that met all of our five requirements to ensure we could implement the LOA framework with available observation data at this study catchment:

- Inclusion of multiple TIM structures (calculated on distributed grid)
- Mass-conserving dynamic glacier evolution model
- Inclusion of multiple linear reservoir runoff-routing routines with dynamic HRUs
- Dynamic temperature lapse rates with on-ice temperature correction.
- Ability to run on HPC

While many codes include one or several of these, none (to our knowledge) include all of them. One option of course, would have been to use multiple model codes, but a very important problem with this (ignoring time requirements) stems from the fact that when it comes to interpreting differences between model acceptability, it is very difficult to determine what aspect of the model brought this about (e.g. was it difference in resolution, different in climate interpolation strategy, difference in time stepping etc.) Other considerations which led us to our own code was:

- Past experience of modifying open source code and potential to introduce errors.
- Ease of extracting required outputs for signature comparison.
- Runtime (many hydrological codes include extra processes that increase runtime).

Driving climate data: The Icelandic Met Office (IMO) provides gridded reanalysis data (P, T and I). They are continuously updating their gridded data sets. Accordingly, I would recommend using the official data set of the IMO, rather than reproducing something that has been investigated for many years by the IMO.

We maintained close contact with the IMO to ensure that we were aware of their most up-to-date climate datasets. In fact, we used their latest ICRA gridded reanalysis data for precipitation (see methodology section). Therefore, we are not entirely sure what you mean by 'official' dataset. Consequently, on receiving this comment we contacted the IMO again who assured us the ICRA dataset is the best available gridded precipitation product currently available (better than say the

Chrochet *et al.* (2007) data which only extends to 2007). For reassurance of this superiority, please see section 6 of the Nawri 2017 referenced in the manuscript.

While we could also have used reanalysis data for temperature and incident solar radiation, we decided to use available observation data taken by the AWS in the catchment and from any nearby IMO weather stations primarily because the reanalysis data is constrained by a sparse set of met stations that are almost entirely located at low altitudes (see Chrochet 2011 for distribution of weather stations that measure temperature in Iceland – I is even lower density). Most importantly, the catchment AWS provide vital information on temperature lapse rates and their variability which are highly catchment-specific and important for simulating ice and snow melt e.g. see Gardner and Sharp (2009) for one example showing this. For these reasons, we preference observation data obtained within the study catchment and from nearby weather stations as opposed to reanalysis data constrained on a sparse network of weather stations.

Figure 3: It is impossible to assess how well the correction worked based on the presented data; see my previous comment. I recommend computing some relevant statistical values (see some of the cited literature).

Yes, on reflection we completely agree that we have not provided adequate information on potential weaknesses of the precipitation data as also noted by referee #2, although we did include $R^2$ values within the text. Therefore, **for the revised manuscript we will include a table of statistics for the precipitation data include seasonal means, standard deviation, coefficient of variation and skewness for the observed and simulated data.**

The described table of statistics has now been included in the mark-up document (see table 1, P12 in mark-up document). We have also included additional text in section 2.4.1 (P13, L21 onwards in mark-up document) which describes the relative strengths and weaknesses of the bias-correction procedure and the precipitation data. Note, we then draw upon these in the discussion section in response to the comments from referee #2.

Fig 4,5,67: please see my concern regarding aggregated monthly ´ or inter-annual comparison of observed and simulated runoff and snow cover; What is the use of modelling daily time steps if only mean monthly results are evaluated? If the objective is only the get monthly means correctly than I recommend using a monthly time step in the modelling framework.

Yes we have somewhat covered this in our comments above, but to clarify, the model runs on an hourly time step and not a daily time step (see methodology section). Also, note that the river discharge signatures characterise model behaviour from monthly to hourly timescales (see table 1) and so are completely dependent on us running the model at a hourly resolution. We hope that makes it clear why we're running it on an hourly time step.

Fig 8: see my comment regarding the evaluation metrics

Discussed in previous responses.

Fig 9: I recommend making a table rather than a figure

We appreciate your preference for using a table rather than a diagram, however, having experimented with using a table previously we found the diagram to be much clearer to the reader, particularly because it allows the inclusion of two numerical values in each cell (one large colour-coded value and one smaller value confined to the top corner of the cell). We found including both

of these in a table (similar in size to table 1) became cluttered and was not clear to the reader. We thank you for the recommendation, but we would prefer to keep these results in diagram format.

Fig ´ 10: why select a case study which has only two glacier volume observations and 23 year of data gap between the two observations? There are numerous case studies which have bi-annual glacier mass balances, which would be a lot more valuable to establish a framework of acceptability for glacio-hydrological models; If the authors want to establish a transparent framework, I recommend selecting a case study with enough observational data to test the framework thoroughly.

We appreciate that there are other case studies with more observation data and there are also countless case studies with less observation data than we have. However, the amount of observation data by no means precludes the use of the LOA framework any more than it precludes the use of more traditional methods of model evaluation. That being said, we do agree that we would like to see the method applied to sites with even more observation data and better constrained driving climate data and we do make this clear in the discussion. Nevertheless, given the general pattern of data scarcity of mountainous regions, we feel this study region is as good as any for testing such a framework.

Fig 11: in my opinion these results suggest that the framework does not work; none of the simulations are acceptable, even according to the standards set by the authors.

We have taken care to address this in our previous responses.

Fig 12: this figure ´ illustrated the discrepancy between observed and simulated snow cover: In my opinion a acceptability framework should be able to identify if a model works during snow intensive and snow poor years. Here we only see that it never really works.

Again, we have taken care to address this in detail above. We would like to reiterate that none of the tested models were deemed acceptable across *all of the signatures*. This does not mean the framework did not work.

Final remark: I do think that this study can become an important contribution to hydrological modelling. However, all of the comments above would need to be accounted for and/or addressed. I would like to encourage the authors to have a thorough discussion on the purpose of this study. Furthermore, I encourage the authors to read some of the literature that is already referenced; many of the concerns addressed above have already been investigated and solutions have been presented. I would like to wish the authors lots of success and good luck with further research on such a framework.

We appreciate this extremely thorough set of referee comments.

In addition to all of the revisions noted above, we also made an additional minor revision after re-reading the manuscript (P4 L20 in the mark-up document). We noted that the sentence, "Such an approach was first proposed by Beven (2006), where observation data uncertainty could be used to define quantitative `limits of acceptability' (LOA) around model evaluation metrics" was making the same point as P3 L23. Accordingly, this has been removed and re-worded.

[revised manuscript text omitted]